Non-dinosaurian dinosauromorphs from the Chinle Formation (Upper Triassic) of the Eagle Basin, northern Colorado: Dromomeron romeri (Lagerpetidae) and a new taxon, Kwanasaurus williamparkeri (Silesauridae)

Martz Jeffrey W. 1 2 martzj@uhd.edu
Small Bryan J. 3
1 Department of Natural Sciences, University of Houston–Downtown , Houston, TX , USA
2 Denver Museum of Nature and Science, Department of Earth Sciences , Denver, CO , USA
3 The Museum of Texas Tech University , Lubbock, TX , USA
Sues Hans-Dieter
Electronic publication date: 2019 Sep 3
Publication date: 2019
Volume: 7
Electronic Location ID: e7551
Received 2019 Apr 24; Accepted 2019 Jul 25
Copyright: © 2019 Martz and Small
Copyright year: 2019
Copyright holder: Martz and Small
License: This is an open access article distributed under the terms of the Creative Commons Attribution License, which permits unrestricted use, distribution, reproduction and adaptation in any medium and for any purpose provided that it is properly attributed. For attribution, the original author(s), title, publication source (PeerJ) and either DOI or URL of the article must be cited.
License URL: https://creativecommons.org/licenses/by/4.0/

Keywords: Eagle Basin, Chinle formation, Lagerpetidae, Silesauridae, Dromomeron, Kwanasaurus, Dinosauromorpha, Dinosauriformes, Triassic

Funding: Denver Museum of Nature and Science and Robert and Cyndi Douglass Financial assistance for field support for this project was provided by the Denver Museum of Nature and Science and Robert and Cyndi Douglass. The funders had no role in study design, data collection and analysis, decision to publish, or preparation of the manuscript.

==============================
The “red siltstone” member of the Upper Triassic Chinle Formation in the Eagle Basin of Colorado contains a diverse assemblage of dinosauromorphs falling outside of Dinosauria. This assemblage is the northernmost known occurrence of non-dinosaurian dinosauromorphs in North America, and probably falls within the Revueltian land vertebrate estimated biochronozone (215–207 Ma, middle to late Norian). Lagerpetids are represented by proximal femora and a humerus referable to Dromomeron romeri. Silesaurids (non-dinosaurian dinosauriforms) are the most commonly recovered dinosauromorph elements, consisting of dentaries, maxillae, isolated teeth, humeri, illia, femora, and possibly a scapula and tibiae. These elements represent a new silesaurid, Kwanasaurus williamparkeri, gen. et sp. nov., which possesses several autapomorphies: a short, very robust maxilla with a broad ascending process, a massive ventromedial process, a complex articular surface for the lacrimal and jugal, and 12 teeth; 14 dentary teeth; an ilium with an elongate and blade-like preacetabular process and concave acetabular margin; a femur with an extremely thin medial distal condyle and a depression on the distal end anterior to the crista tibiofibularis. The recognition of K. williamparkeri further demonstrates the predominantly Late Triassic diversity and widespread geographic distribution across Pangea of the sister clade to Asilisaurus, here named Sulcimentisauria. Silesaurid dentition suggests a variety of dietary specializations from faunivory and omnivory in the Middle Triassic and early Late Triassic (Carnian), to herbivory in the Late Triassic (Carnian and Norian), with the latter specialization possibly coinciding with the radiation of Sulcimentisauria across Pangea. The extremely robust maxilla and folidont teeth of K. williamparkei may represent a strong herbivorous dietary specialization among silesaurids.

Introduction

By the final years of the 20th century, the diversity of dinosauromorphs across Pangea was thought to follow a simple pattern during the Triassic Period. The non-dinosaurian dinosauromorphs were restricted to the Middle Triassic of South America (Sereno & Arcucci, 1994a, 1994b), and Dinosauria was restricted to the Late Triassic, with theropods, sauropodomorphs, and ornithischians all having a global distribution that included western North America (Hunt & Lucas, 1994; Long & Murry, 1995; Padian & May, 1993).

This picture began to change drastically in the 21st century with the description of Silesaurus opolensis (Dzik, 2003) from the Carnian or Norian Krasiejów beds of Poland, which revealed that non-dinosaurian dinosauriforms survived into the Late Triassic. This prompted an extensive re-evaluation of the record of putative dinosaur fossils from the Upper Triassic Chinle Formation of New Mexico and Arizona, and the equivalent Dockum Group of Texas. This work revealed a previously unrecognized diversity of non-dinosaurian dinosauromorphs surviving into the Late Triassic of North America (Ezcurra, 2006; Nesbitt et al., 2009a; Nesbitt & Chatterjee, 2008; Irmis et al., 2007a; Martz et al., 2013; Sarigül, 2016) as well as that ornithischians and sauropodomorphs were probably absent in North America prior to the Jurassic (Nesbitt, Irmis & Parker, 2007; Irmis et al., 2007b).

The description of the lagerpetid dinosauromorphs Dromomeron romeri Irmis et al., 2007a and Dromomeron gregorii Nesbitt et al., 2009a from the Chinle Formation and Dockum Group of western North America extended the record of the Lagerpetidae from South America into the Norian stage of the Late Triassic of North America (Irmis et al., 2007a; Nesbitt, Irmis & Parker, 2007; Marsh, 2018). The Chañares Formation, which produced Lagerpeton chanarensis, was originally thought to be Middle Triassic (Romer, 1971) but has recently been radioisotopically dated as early Carnian (Marsicano et al., 2016), indicating that known lagerpetids were restricted to the Late Triassic of North America and South America (Müller, Langer & Dias-da-Silva, 2018).

The taxa Eucoelophysis baldwini Sullivan & Lucas, 1999 from the Chinle Formation of New Mexico (Ezcurra, 2006; Nesbitt, Irmis & Parker, 2007; Irmis et al., 2007a; Breeden et al., 2017), as well as Technosaurus smalli Chatterjee, 1984 and Soumyasaurus aenigmaticus Sarigül, Agnolin & Chatterjee, 2018 from the Dockum Group of Texas, demonstrate that silesaurids also occurred in North America during the Late Triassic (Nesbitt, Irmis & Parker, 2007; Martz et al., 2013). Additional discoveries give silesaurids a global record spanning the Middle to Late Triassic of both Gondwana and Laurasia (Langer et al., 2013; Martinez et al., 2015; Peecook et al., 2013, 2017). Both lagerpetids and silesaurids coexisted with dinosaurs in Gondwana at least as early the late Carnian (Martinez et al., 2012; Garcia et al., 2019), and in both Gondwana and Laurasia at least as late as the late Norian (Langer & Ferigolo, 2013; Marsh, 2018).

The Eagle Basin of Colorado (Fig. 1A) contains some of the northernmost exposures of the Chinle Formation (Poole & Stewart, 1964; Dubiel, 1992), a unit that has been studied more extensively in the Colorado Plateau (Stewart, Poole & Wilson, 1972; Blakey & Gubitosa, 1983; Lucas, 1993; Dubiel, 1994; Martz et al., 2017). During the Late Triassic, the Eagle Basin was separated from the Colorado Plateau depocenter by the Ancestral Front Range and Ancestral Uncompahgre Highlands (Dubiel, 1992, 1994). Over 20 years of collection from Eagle Basin localities by the junior author has yielded an abundance of vertebrate fossils, mostly consisting of isolated elements (Small & Sedlmayr, 1995; Small, 2001, 2009; Martz, Mueller & Small, 2003; Small & Martz, 2013; Martz & Small, 2016; Pardo, Small & Huttenlocker, 2017), that include rare fish, the stem caecilian Chinlestegophis jenkinsi (Pardo, Small & Huttenlocker, 2017), a possible metoposaurid, a leptopleuronine procolophonid similar to Libognathus Small, 1997, a variety of small diapsids, rare phytosaur elements that cannot be assigned to alpha taxa, the aetosaur Stenomyti huangae Small & Martz, 2013, another aetosaur that may be referable to Rioarribasuchus Lucas, Hunt & Spielmann, 2006, shuvosaurids, rauisuchids, crocodylomorphs, and dinosauromorphs. A variety of plant macrofossils have also been recovered from the area (BJ Small & JW Martz, 2013, personal observations).

Figure 1 Chinle Formation exposures in the Eagle Basin of northern Colorado.

(A) Map of Colorado showing approximate location of localities. (B) Stratigraphic section of the Chinle Formation showing approximate stratigraphic interval of dinosauromorph localities (modified from Derby Junction section of Dubiel, 1992: fig. 4). (C) Exposures of the red siltstone member along the Colorado River north of I-70 at 13S 033415 4412881 NAD 27 showing the approximate division between the coarser facies similar to the Petrified Forest Member and the finer-grained facies similar to the Owl Rock Member. (D) Bone preserved in fine-grained silty to very fine-grained sandstone. (E) Intrabasinal conglomerate beds that have produced the bulk of the specimens.

Here, we describe the first occurrence of the lagerpetid Dromomeron romeri from the Chinle Formation of the Eagle Basin of Colorado, which represents the northernmost occurrence of the genus, and a new genus and species of silesaurid, Kwanasaurus williamparkeri. This new taxon is based primarily on isolated elements (Table 1) exhibiting a distinctive suite of derived characters not recognized in any other silesaurid. Kwanasaurus is the fourth silesaurid alpha taxon recognized from North America, and the northernmost silesaurid known from the Americas. Material from the Eagle Basin localities referable to Neotheropoda (Small, 2009) will be described in detail elsewhere.

Table 1 Basal dinosauromorph specimens.

Taxon	Specimen #	Element	Locality	
Dromomeron romeri	DMNH EPV.54826 (voucher)	Proximal left femur	DMNH 1306 (Main Elk Creek)	
DMNH EPV.29956	Complete right humerus	DMNH 1306 (Main Elk Creek)	
DMNH EPV.63873	Proximal right femur	DMNH 1306 (Main Elk Creek)	
Dinosauriformes	DMNH EPV.67956	Partial right scapula	DMNH 3980 (Lost Bob)	
DMNH EPV.27699	Worn proximal left femur	DMNH 1306 (Main Elk Creek)	
DMNH EPV.43126	Worn proximal left femur	DMNH 1306 (Main Elk Creek)	
DMNH EPV.43588	Worn proximal left femur	DMNH 1306 (Main Elk Creek)	
DMNH EPV.44616	Worn proximal left femur	DMNH 1306 (Main Elk Creek)	
DMNH EPV.63875	Complete right tibia	DMNH 4629 (Lost Bob East)	
DMNH EPV.63872	Proximal right tibia	DMNH 3980 (Lost Bob)	
DMNH EPV.56652	Worn proximal tibia	DMNH 1306 (Main Elk Creek)	
DMNH EPV.67955	Proximal left tibia	DMNH 3980 (Lost Bob)	
Kwanasaurus parkeri	DMNH EPV.65879 (holotype)	Partial left maxilla	DMNH 4340 (Burrow Cliff)	
DMNH EPV.63650	Partial right maxilla	DMNH 3980 (Lost Bob)	
DMNH EPV.125921	Partial left maxilla	DMNH 4629 (Lost Bob East)	
DMNH EPV.125923	Partial right maxilla	DMNH 4629 (Lost Bob East)	
DMNH EPV.63136	Nearly complete left dentary	DMNH 3980 (Lost Bob)	
DMNH EPV.63135	Partial right dentary	DMNH 3980 (Lost Bob)	
DMNH EPV.63660	Left anterior dentary	DMNH 3980 (Lost Bob)	
DMNH EPV.65878	Partial left dentary	DMNH 4629 (Lost Bob East)	
DMNH EPV.57599	Partial right? dentary	DMNH 1306 (Main Elk Creek) South 6	
DMNH EPV.43577	Tooth	DMNH 1306 (Main Elk Creek) South 2	
DMNH EPV.63142	Tooth	DMNH 3980 (Lost Bob)	
DMNH EPV.63143	Tooth	DMNH 3980 (Lost Bob)	
DMNH EPV.63661	Tooth	DMNH 3980 (Lost Bob)	
DMNH EPV.125922	Tooth	DMNH 4629 (Lost Bob East)	
DMNH EPV.59302	Nearly complete left humerus	DMNH 1306 (Main Elk Creek) South 7	
DMNH EPV.48506	Complete left ilium	DMNH 1306 (Main Elk Creek)	
DMNH EPV.63653	Nearly complete left ilium	DMNH 3980 (Lost Bob)	
DMNH EPV.52195	Partial ilium	DMNH 1306 (Main Elk Creek) South	
DMNH EPV.34579	Nearly complete femur	DMNH 692 (Derby Junction)	
DMNH EPV.54828	Proximal right femur	DMNH 3492 (Shuvosaur Surprise)	
DMNH EPV.59311	Proximal right femur	DMNH 3492 (Shuvosaur Surprise)	
DMNH EPV.44616	Proximal right femur	DMNH 1306 (Main Elk Creek) North 2	
DMNH EPV.56651	Proximal left femur	DMNH 1306 (Main Elk Creek)	
DMNH EPV.59301	Proximal left femur	DMNH 1306 (Main Elk Creek) South	
DMNH EPV.63139	Proximal left femur	DMNH 3980 (Lost Bob)	
DMNH EPV.63874	Proximal left femur	DMNH 4629 (Lost Bob East)	
DMNH EPV.125924	Proximal right femur	DMNH 4629 (Lost Bob East)	
Silesauridae?	DMNH EPV.34028	Distal right femur	DMNH 1306 (Main Elk Creek)	
DMNH EPV.59310	Distal right femur	DMNH 3492 (Shuvosaur Surprise)	
Note:

Voucher specimens are indicated in boldface; the voucher specimen for Kwanasaurus williamparkeri (DMNH EPV.65879) serves as voucher specimen for both Dinosauriformes and Silesauridae.

Geologic Setting

The fossils that are the focus of this study come from the middle of the informally named “red siltstone member” of the Chinle Formation (Figs. 1B–1E), a 100–150 m section of steep, bench forming red beds that overlie the Gartra Member, a conglomeratic sandstone considered to form the base of the Chinle Formation. The Eagle Basin Chinle Formation unconformably overlies the Permian Maroon Formation and Early Triassic State Bridge Formation, and is unconformably overlain by the Early Jurassic Entrada Formation (Poole & Stewart, 1964; Stewart, Poole & Wilson, 1972; Dubiel & Skipp, 1989; Dubiel, 1992).

The red siltstone member contains sandstones and conglomerate lenses interbedded with siltstones and very fine sandstones showing abundant evidence of pedogenic modification; these beds have been interpreted as moderate to high sinuosity channel sandstones and overbank deposits (Dubiel, 1992). The red siltstone member shows a subtle fining upward sequence in which the upper part of the sequence is almost entirely siltstone to very fine-grained sandstone with more evidence of pedogenic development than seen in the lower part of the member (Fig. 1B; JW Martz & BJ Small, 2016, personal observations). Although Poole & Stewart (1964) correlated the red siltstone member with the Church Rock Member of Utah, the sedimentological transition from the lower to upper red siltstone member (Figs. 1B–1C) resembles the shift from the Petrified Forest Member to the Owl Rock Member in the Colorado Plateau (Blakey & Gubitosa, 1983; Martz et al., 2017). However, the current authors have not pursued sufficiently detailed lithostratigraphic correlations between the Eagle Basin and the Colorado Plateau to resolve the precise relationships between these units.

Vertebrate specimens from the Eagle Basin have primarily been recovered from the lower half of the red siltstone member, 50–60 m below the top of the Chinle Formation, in the coarser-grained “Petrified Forest-like” facies (Figs. 1B–1C). Specimens have been recovered from the highly productive Main Elk Creek locality near Newcastle, Colorado (DMNH loc. 1306), as well as the Derby Junction (DMNH loc. 692; Dubiel, 1992, p. W16), Lost Bob (DMNH loc. 3980), Lost Bob East (DMNH loc. 4629), Burrow Cliff (DMNH loc. 4340) and Shuvosaur Surprise (DMNH loc. 3492) localities. These localities all occur in a narrow stratigraphic interval near Derby Junction, Colorado (Fig. 1B). Specimens consist mostly of isolated bones, with occasional associated remains and rare articulated elements, recovered from small conglomeratic lenses (Fig. 1E) probably representing small channels transporting remains under high energy conditions (BJ Small & JW Martz, 2013, personal observations). The finer-grained overbank siltstones (Fig. 1D) represent lower energy conditions and have yielded some of the best-articulated material (e.g., the holotype of Stenomyti huangae Small & Martz, 2013).

The precise age of the Eagle Basin Chinle localities is difficult to determine, as these strata have not yet yielded a diagnostic palynoflora, phytosaur cranial material, or radioisotopic dates required for definitive biostratigraphic or chronostratigraphic correlations with the better-calibrated Chinle Formation of the Colorado Plateau and Dockum Group of the southern High Plains (Irmis et al., 2011; Ramezani, Fastovsky & Bowring, 2014; Martz & Parker, 2017). However, specimens possibly referable to the leptopleuronine procolophonid Libognathus (DMNH EPV.56657), the aetosaur Rioarribosuchus (e.g., DMNH EPV.48018, 48019), and the lagerpetid Dromomeron romeri (DMNH EPV.54826) (Small, 2009; Small & Martz, 2013) all provide circumstantial evidence that the fossil localities fall within the Revueltian estimated biochronozone (sensu Martz & Parker, 2017) which is probably Alaunian to Sevatian (middle to late Norian, 215–207 Ma), although Dromomeron romeri also occurs in the Apachean estimated biochronozone (late Norian to Rhaetian, 207–202 Ma) (Marsh, 2018). Moreover, the aetosaur Stenomyti huangae (Small & Martz, 2013) is very similar to Aetosaurus material from European strata that are probably also Norian (Wild, 1989; Heckert & Lucas, 2000; Bachmann & Kozur, 2004), and Aetosaurus-like osteoderms have been identified from the Revueltian and Apachean estimated biochronozones elsewhere in the western United States (Lucas, 1998; Heckert et al., 2007; Martz, 2008).

Methodology

All material described below from the Main Elk Creek, Lost Bob, Shuvosaur Surprise, Burrow Cliff, and Derby Junction localities are isolated and associated elements from larger bone assemblages. We rely primarily on a synapomorphy-based approach for identification of vertebrates from the Eagle Basin localities following the framework established for other Upper Triassic localities (Nesbitt & Stocker, 2008; Martz et al., 2013). This testable approach utilizes the presence of discrete apomorphies in a phylogenetic framework to determine the taxonomic placement of individual specimens (Bever, 2005; Bell, Gauthier & Bever, 2010). Incomplete specimens lacking clear apomorphies may in some cases be tentatively assigned to particular taxa based on close association or similarity with more complete specimens possessing apomorphies. Moreover, we have designated voucher specimens for all identified taxa, which are usually the most complete or best-preserved specimens (Table 1). Measurements for selected appendicular elements are given in Table S1, illustrated in Fig. S1, and described in Appendix 1.

The electronic version of this article in Portable Document Format will represent a published work according to the International Commission on Zoological Nomenclature (ICZN), and hence the new names contained in the electronic version are effectively published under that Code from the electronic edition alone. This published work and the nomenclatural acts it contains have been registered in ZooBank, the online registration system for the ICZN. The ZooBank Life Science Identifiers (LSIDs) can be resolved and the associated information viewed through any standard web browser by appending the LSID to the prefix http://zoobank.org/. The LSID for this publication is: urn:lsid:zoobank.org:pub:20FCEEA6-4512-42FD-BAE9-A570BF4611F4. The online version of this work is archived and available from the following digital repositories: PeerJ, PubMed Central and CLOCKSS.

Systematic Paleontology

Dinosauromorpha Benton, 1985 sensu Sereno, 1991

Lagerpetidae Arcucci, 1986 sensu Nesbitt et al., 2009a

Dromomeron Irmis et al., 2007a

Dromomeron romeri Nesbitt, Irmis & Parker, 2007

Referred specimens. DMNH EPV.54826 (Fig. 2), proximal left femur (voucher specimen); DMNH EPV.63873 (Fig. 3), proximal right femur (and other associated elements, at least some of which are pseudosuchians and therefore not part of the same individual); DMNH EPV.29956 (Fig. 4), right humerus.

Figure 2 Dromomeron romeri voucher specimen (DMNH EPV.54826), proximal left femur, stereopairs, and interpretive drawings.

(A) Proximal view, (B) anterolateral view, (C) anteromedial view, (D) posteromedial view, (E) posterolateral view. See text for abbreviations. Scale bar = 2 cm.

Figure 3 Dromomeron romeri (DMNH EPV.63873), proximal right femur, labeled steropairs.

(A) Anterolateral view, (B) anteromedial view, (C) posteromedial view, (D) posterolateral view. See text for abbreviations. Scale bar = 1 cm.

Figure 4 Dromomeron romeri (DMNH EPV.29956), right humerus, labeled stereopairs.

(A) Proximal view, (B) anterior view, (C) medial view, (D) posterior view, (E) lateral view, (F) proximal view showing angle of torsion between long axes of proximal and distal ends, gray lines represent the long axes of the proximal and distal ends. See text for abbreviations. Scale bar = 2 cm.

Description and discussion

Femur

Two proximal femora (Figs. 2–3; DMNH EPV.54826; DMNH EPV.63873) recovered from Main Elk Creek possess several apomorphies of the lagerpetid Dromomeron (Irmis et al., 2007a; Nesbitt et al., 2009a; Langer et al., 2013). The femoral heads are distinctly hook-shaped with a ventrolateral emargination (in Figs. 2–3) as in Dromomeron romeri, Lagerpeton chanarensis, and Ixalerpeton polesinensis (Nesbitt et al., 2009a; Cabreira et al., 2016) and a well-developed posteromedial tuber (pmt in Figs. 2–3) that is much larger than the anteromedial tuber (amt in Figs. 2–3), which is barely discernible (synapomorphies of Lagerpetidae; Nesbitt et al., 2009a). The proximal ends of the femora form the smooth arc characteristic of lagerpetids, with the facies articularis antitrochanterica (faa in Figs. 2–3) extending more distally on the posteromedial side of the proximal femur as in other dinosauromorphs (Nesbitt et al., 2009a). An anterolateral tuber is absent so that the lateral side of the proximal femur head is relatively flattened in DMNH EPV.54826 (Fig. 2B), a feature shared by lagerpetids and shuvosaurids (Nesbitt, 2011), although the region is nonetheless somewhat swollen in DMNH EPV.63873. There is no indication of the roughened anterior trochanter or posteromedial muscle scar diagnostic of Dromomeron gigas (Martinez et al., 2015). The anterolateral edge of the proximal end of the femora is sharper than the posteromedial edge of the proximal end, although it does not form the distinct dorsolateral trochanter present in dinosauriforms (Nesbitt, 2011: character state 307-0).

Below this sharp edge, the anterolateral surface of the proximal end of the femur in DMNH EPV.54826 is slightly concave (cnc in Fig. 2A), although the region is not fully prepared in DMNH EPV.63873. This concavity distinguishes Dromomeron from Lagerpeton, in which the anterolateral surface is flattened (Nesbitt et al., 2009a, p. 502). At least in DMNH EPV.54826, where some of the shaft is preserved, both lesser (anterior) and fourth trochanters are completely absent (autapomorphies of Dromomeron romeri; Nesbitt et al., 2009a). The posteromedial surface of the femur shaft is flattened and a scar for M. caudifemoralis longus cannot be clearly discerned (Fig. 2D), while the anterolateral surface of the shaft is more convex (cnv in Fig. 2B).

Humerus

The only previously published non-dinosauriform dinosauromorph humerus is for Ixalerpeton, which was figured but not described in detail (Cabreira et al., 2016: fig. 1F) and a passing mention by Nesbitt (2011: p. 125) of a humerus he assigned to Dromomeron gregorii (TMM 31000-1329) without description. A slender right humerus (DMNH EPV.29956; Fig. 4) from the Main Elk Creek locality may also belong to Dromomeron.

The proximal end and deltopectoral crest of DMNH EPV.29956 (dc in Fig. 4) are strongly mediolaterally expanded relative to the shaft as in most archosauriforms, including Ixalerpeton (Cabreira et al., 2016) and the dinosauriforms Asilisaurus, Lewisuchus, and Marasuchus (Langer et al., 2013). The proximal end and deltopectoral crest are both much less expanded in the derived silesaurids Silesaurus and Diodorus, as well as in shuvosaurids (Dzik, 2003; Nesbitt, 2011; Kammerer, Nesbitt & Shubin, 2012; Langer et al., 2013).

The expanded proximal part of the humerus is medially inclined (Figs. 4B and 4D). The proximal end bears two distinct swellings, possibly the ectotuberosity and entotuberosity of Welles (1984) (ec and en in Figs. 4A–4B and 4D), and a pointed medial or internal tuberosity (mt in Fig. 4). The medial tuberosity is slightly displaced distally relative to the proximal edge of the head as in most dinosauromorphs including Ixalerpeton (Cabreira et al., 2016: fig. 1F), but not in Silesaurus (Dzik, 2003: fig. 9), and Herrerasaurus (Sereno, 1994: fig. 3), where the medial tuberosity is level with the proximal edge of the humerus.

The deltopectoral crest of DMNS EPV.29956 (dc in Fig. 4) is separated from the proximal end of the humerus by a thin crest of bone (tc in Figs. 4B and 4D–4E) as in dinosaurs (Nesbitt, 2011). However, as with most non-dinosaurian dinosauriforms, the deltopectoral crest retains the plesiomorphic state of being subtriangular with the apex less than a third the length of the bone from the proximal end (Nesbitt, 2011); the deltopectoral crest in dinosaurs is subrectangular and extends more than a third of the length of the humerus from the proximal end (Langer & Benton, 2006; Nesbitt, 2011). The lagerpetid Ixalerpeton differs from most non-dinosaurian dinosauriforms in that the crest also extends more than a third the length of the humerus (Cabreira et al., 2016).

Compared to Marasuchus lilloensis (Bonaparte, 1975: fig. 9), the shaft of the humerus in DMNH EPV.29956 is very slender compared to the distal end, much like Ixalerpeton (Cabreira et al., 2016: fig. 1F). A faintly preserved ectepicondylar flange and groove are present as in phytosaurs and pseudosuchian archosaurs (ecf in Fig. 4B), although these are absent in nearly all ornithodirans (Nesbitt, 2011). However, Nesbitt (2011: p. 125) noted that an ectepicondylar groove was present in the humerus he assigned to Dromomeron gregorii (TMM 31000-1329); whether or not a groove is present in Ixalerpeton polesinensis is unclear (Cabreira et al., 2016: fig. 1F). The ectepicondyle (lateral distal condyle) projects more distally than the entepicondyle (medial condyle (mc)) (ect and ent in Figs. 4B–4E) as it does in Ixalerpeton (Cabreira et al., 2016: fig. 1F). The posterior side of the distal end is deeply concave, with the concavity tapering proximally (cnc in Figs. 4C–4D).

Viewed proximally, the long axes of the distal and proximal ends of the humerus are not parallel, but offset at an angle of about 45° (Fig. 4F). The presence of torsion between the proximal and distal ends of the humerus is variable amongst dinosauromorphs. It is present to at least some extent in Eoraptor lunensis, sauropodomorphs, and most basal theropods (Tykowski, 2005: character state 172-1), but absent (i.e., the long axes of the proximal and distal ends are parallel in proximal view) in Marasuchus, Herrerasaurus, and basal ornithischians (Tykowski, 2005).

Given the presence of a single putative dinosaurian synapomorphy (a thin crest of bone separating the deltopectoral crest form the proximal end, also shared with Ixalerpeton) combined with a plesiomorphy absent in dinosaurs (subtriangular deltopectoral crest that does not extend far down the shaft), and the lack of any apomorphies diagnosing any other archosauriform clade, DMNH EPV.29956 is tentatively assigned to Dromomeron. This humerus is very distinct from those of both dinosaurs and silesaurids (see below).

Dinosauriformes Novas, 1992

Referred specimens. DMNH EPV.67956 (Fig. 5), partial right scapula; several worn proximal left femora (none figured): DMNH EPV.27699, DMNH EPV.43126, and DMNH EPV.43588; DMNH EPV.63875, complete right tibia (Fig. 6), DMNH EPV.56652 (Fig. 7A), worn proximal tibia; DMNH EPV.63872 (Figs. 7B–7F), proximal right tibia; DMNH EPV.67955 (Figs. 7G–7K), proximal left tibia.

Figure 5 Dinosauriformes (DMNH EPV.67956), right scapula, labeled stereopairs.

(A) Anterior view, (B) medial view, (C) posterior view, (D) lateral view, (E) ventral view. Missing areas outlined with dots. See text for abbreviations. Scale bar = 2 cm.

Figure 6 Dinosauriformes (DMNH EPV.63875), right tibia, labeled stereopairs.

(A) Proximal view, (B) anterior view, (C) medial view, (D) posterior view, (E) lateral view, (F) distal view. See text for abbreviations. Scale bar = 2 cm.

Figure 7 Dinosauriformes tibiae.

(A) DMNH EPV.56652, worn proximal tibia in lateral view. DMNH EPV.67955, proximal end of right tibia in (B) proximal view. (C) Anterior view, (D) medial view, (E) posterior view, (F) lateral view. DMNH EPV.67955, proximal left tibia stereopairs in (G) proximal view, (H) anterior view, (I) medial view, (J) posterior view, (K) lateral view. See text for abbreviations. Scale bar = 2 cm.

Description and discussion. Some elements in the Eagle Basin collection possess dinosauriform apomorphies but cannot be assigned with certainty to a more specific group. These elements are consistent with either silesaurids or basal (non-neotheropod) theropods (Nesbitt et al., 2009b), but lack apomorphies that would allow them to be assigned definitively to either group. They are discussed here as potential silesaurid elements.

Scapula

DMNH EPV.67956 (Fig. 5) is a mostly complete right scapula from Lost Bob missing much of the ventral anterior edge and the dorsal apex. The scapula is mediolaterally thickest ventrally at the articular glenoid (ag in Figs. 5C and 5E), and thins dorsally. The posteroventrally-facing surface of the glenoid is ovate, slightly concave, surfaced with spongy bone, and projects somewhat posterolaterally (Fig. 5E). Anterior to the glenoid, the scapula forms a subtriangular articular surface for the coracoid (co.ar in Figs. 5B and 5E). Immediately above the glenoid, where the shaft is thickest, the posterior margin of the scapula is flattened (Fig. 5C), the medial margin is slightly concave (cnc in Fig. 5B), and the lateral margin is slightly convex (cnv in Fig. 5D). The anterior part of the scapula prominence, including the preglenoid fossa, is not preserved except for part of the sharp-edged, posterodorsally-sloping, thin crest connecting the dorsal edge of the prominence to the anterior side of the shaft (tc in Figs. 5A–5B and 5D). The absence of the scapula prominence is unfortunate, as the size of the ridge bordering the preglenoid fossa dorsally is much more sharper and narrower in at least some silesaurids compared to dinosaurs (Langer & Ferigolo, 2013), and may allow the two clades to be distinguished.

The anterior and posterior edges of the scapula shaft diverge slightly dorsally, indicating a widened dorsal apex, although only a small part of the apex is preserved (asc in Fig. 5). However, it is evident that the blade length of the element is more than three times its dorsal width. Such “strap-like” scapulae occur in silesaurids and neotheropods (Nesbitt, 2011: character state 218-1), but also in Tawa hallae (Nesbitt et al., 2009b: fig. 2B). The lateral surface of the scapula shaft is convex and the medial surface is slightly more flattened. Both surfaces are covered with faint longitudinal striations. The anterior edge of the shaft is also somewhat sharper than the posterior edge, and becomes very sharp as the shaft thins approaching the apex (Fig. 5A). The preserved part of the dorsal apex thins very abruptly (best seen in Fig. 5C). This may indicate that an ossified suprascapula was present. Two tiny elongate depressions just below this abrupt thinning on the medial surface seem to be natural, and may end in tiny foramina.

The overall long and slender form of the scapula compares well with Silesaurus (Dzik, 2003: fig. 9), Sacisaurus (Langer & Ferigolo, 2013: fig. 8I), and the basal theropod Tawa (Nesbitt et al., 2009b: fig. 2B). In most Late Triassic and Early Jurassic theropods, the element seems to be somewhat shorter with a much broader dorsal apex (Rowe, 1989: fig. 2; Colbert, 1989: figs. 2–3; Carpenter, 1997: fig. 5; Sereno, 1994; Tykowski, 2005: figs. 59–62; Langer, Bittencourt & Schultz, 2011; Martinez et al., 2011). However, in the absence of known silesaurid apomorphies, the Eagle Basin scapula can only be assigned with certainty to Dinosauriformes.

Femur

Several un-figured proximal femora (DMNH EPV.27699, DMNH EPV.43126, DMNH EPV.43588, and DMNH EPV.44616), are known from Main Elk Creek that are referable to Dinosauriformes based on the presence of an anterior trochanter but lack of a trochanteric shelf; moreover, DMNH EPV.43126 possesses a posterolateral trochanter, which also diagnoses Dinosauriformes (Langer & Benton, 2006; Nesbitt, 2011). Preserved portions of these elements are identical to the silesaurid femora described below, and therefore likely belong to Kwanasaurus, but the proximal ends are too badly worn to preserve critical silesaurid apomorphies. As a result, they can only be assigned to Dinosauriformes.

Tibia

DMNH EPV.63875 (Fig. 6), a complete right tibia from Lost Bob East, DMNH EPV.56652 (Fig. 7A), a badly worn proximal tibia from Main Elk Creek, DMNH EPV.63872 (Figs. 7B–7F), a proximal right tibia from Lost Bob, and DMNH EPV.67955 (Figs. 7G–7K), a proximal left tibia from Lost Bob, can also be referred to Dinosauriformes. The combination of character states in these elements is consistent with silesaurids, although specific silesaurids synapomorphies cannot be identified.

The proximal ends of the tibiae possess several important apomorphies. The posterior edges of the lateral and medial condyles at the proximal end (lc and mc in Figs. 6–7) are adjacent in all specimens except for DMNH EPV.56652 (Fig. 7A), which is too badly worn to determine if it shares this condition. Adjacent proximal condyles occur in silesaurids and theropods (Langer & Benton, 2006; Nesbitt et al., 2009b; Nesbitt, 2011). However, the proximal surfaces of DMNH EPV.63875 and DMNH EPV.63872 are gently convex (Figs. 6C, 6E and 7D), and the cnemial crest is nearly straight (cc in Figs. 6–7), as in non-dinosaurian dinosauromorphs. Moreover, unlike the condition in neotheropods, the cnemial crest does not project more proximally than the rest of the proximal end, and is not separated from condyles by a concavity (Figs. 6C, 6E, 7D, 7F, 7I and 7K). A distinct ridge is also present on the lateral side of the cnemial crest DMNH EPV.63875 and DMNH EPV.67955 (where cc is labeled in Figs. 6E and 7K). Unlike the basal theropod Chindesaurus bryansmalli (Long & Murry, 1995; Nesbitt et al., 2009b; Marsh et al., 2016), the lateral and mcs are about the same size (Figs. 6A and 7B). The posteromedial surface of the proximal end of the tibiae has a distinct swelling adjacent to the mc in DMNH EPV.63875 (sw in Fig. 6) that is apparently absent in the smaller specimens. All specimens possess a distinct fibular crest (fc in Figs. 6E, 7A, 7F and 7K) as in most Triassic dinosauriforms except for Tawa (Nesbitt et al., 2009b). The fibular crest extends parallel to the long axis of all tibiae and terminates distally before reaching the midpoint of the element.

The shafts of the tibiae are mediolaterally somewhat constricted and oval in cross section for about the proximal third, then becoming subcircular in cross section by the midpoint of the shaft. Roughly the distal third of the posterolateral edge of the shaft of DMNH EPV.63875 is slightly constricted above the posterolateral flange of the distal end (Fig. 6D).

In DMNH EPV.63875, the distal end of the tibia bears a distinct slightly distally projecting and blade-like posterolateral process (plp in Figs. 6D–6F) as in other dinosauriforms. This seems to be more similar to the pronounced crest-like posterolateral process of Sacisaurus (Langer & Ferigolo, 2013: fig. 18) than to the smaller process of Silesaurus (Dzik, 2003: fig. 13). There is a broad depression for the ascending process of the astragalus (as.ar in Figs. 6E–6F). Immediately anterior to this, the distal end of the tibia is distinctly mediolaterally thicker than the posterolateral process, a character shared by silesaurids and saurischian dinosaurs (Novas, 1996; Langer & Benton, 2006; Nesbitt, 2011). Anterior to the depression for the ascending process, the anterior part of the distal end projects slightly anterior to the tibia shaft as a slightly pinched eminence (Figs. 6E–6F).

These tibiae compare well overall to the element in Silesaurus (Dzik, 2003: fig. 13) and Sacisaurus (Langer & Ferigolo, 2013: fig. 18), and lack character states present in neotheropods such as dorsal expansion of the cnemial crest, a posterolateral concavity at the distal end, and a proximodistally oriented ridge on the posterior side of the distal end (distinct from the posterolateral flange) (Nesbitt, 2011). However, it cannot be completely ruled out that the elements belong to non-neotheropod theropods, as the presence of these characters is variable in basal theropods such as Tawa and herrerasaurids (Nesbitt et al., 2009b: p. 1532; Nesbitt, 2011), and the tibiae of Eodromaeus murphi and Daemonosaurus chauliodus are unknown (Martinez et al., 2011; Sues et al., 2011). However, for reasons discussed above the elements are not referable to Tawa or Chindesaurus.

Silesauridae Nesbitt et al., 2010

Diagnosis. See Appendix 3.

Sulcimentisauria clade nov.

Definition (stem-based). The most inclusive clade that includes Silesaurus opolensis Dzik 2003 but not Asilisaurus kongwe Nesbitt et al. 2010.

Diagnosis. See Appendix 3.

Etymology. Latin sulcus- “grooved” + Latin mentum “chin” + Greek sauros “lizard.” In reference to the ventrally placed Meckelian groove on the dentary.

Kwanasaurus gen. nov.

LSID. urn:lsid:zoobank.org:act:E9514954-F9FD-4D79-A620-D705122D59D5

Type species. Kwanasaurus williamparkeri.

Etymology. Ute kwana- “eagle” + Greek sauros “lizard.” The generic name honors the town and county of Eagle in Colorado, located near the fossil localities that produced the type and referred specimens, as well as the Ute people. The town and county of Eagle are named for the Eagle River (Río Águila in Spanish), said to be translated from a local Ute name for the river or from the name of a Ute chief.

Autapomorphic diagnosis. Kwanasaurus is distinguished from all other silesaurid taxa by the following autapomorphies: Main body and posterior process of maxilla extremely short and robust; ascending process of the maxilla extends at least half the anteroposterior length of the element; prominent posterolateral flange and complex jugal and lacrimal articulations on posterior end of posterior process of the maxilla; massive subtriangular, ventromedially oriented flange on medial surface of the maxilla; 12 maxillary teeth; 14 dentary teeth; ilium with elongate and blade-like preacetabular process that extends beyond the pubic peduncle; concave ventral acetabular margin of ilium; medial condyle at distal end of femur very thin compared to lateral condyle and crista tibiofibularis; depression on distal end of the femur anterior to the crista tibiofibularis.

Differential diagnosis. Aside from autapomorphies, Kwanasaurus possesses the following combination of character states in relation to various silesaurid taxa: Kwanasaurus shares with Lewisuchus/Pseudolagosuchus a tooth row that extends to the posterior end of the maxilla and a fourth trochanter on the femur; Kwanasaurus differs from Lewisuchus/Pseudolagsuchus in having a steeply rising ascending process on the maxilla and in possessing broad and coarsely denticulate folidont teeth. Kwanasaurus shares with Asilisaurus a Meckelian groove that does not extend through the symphysis; Kwanasaurus differs from Asilisaurus in possessing a dentary with a ventrally positioned Meckelian groove, broad and coarsely denticulate folidont teeth, and a strongly “saddle-shaped” ilium. Kwanasaurus shares with the large Manda beds silesaurid an un-notched anterior trochanter. Kwanasaurus shares with Silesaurus a steeply rising ascending process of the maxilla, a dentary with a ventrally positioned Meckelian groove, distinct torsion between the proximal and distal ends of the humerus, a strongly “saddle-shaped” ilium, and a fourth trochanter; Kwanasaurus differs from Silesaurus in having a tooth row that extends to the posterior end of the maxilla, having a dentary with a pronounced lateral ridge, having a dentary with a Meckelian groove that does not extend through the symphysis, and in possessing broad and coarsely denticulate folidont teeth; Kwanasaurus shares with some individuals of Silesaurus an un-noched anterior trochanter and the absence of a trochanteric shelf. Kwanasaurus shares with Sacisaurus a dentary with a ventrally positioned Meckelian groove, broad and coarsely denticulate folidont teeth, and a fourth trochanter on the femur; Kwanasaurus differs from Sacisaurus in having a tooth row that extends to the posterior end of the maxilla, having a pronounced lateral ridge on the dentary, having an anterolateral groove on the dentary that extends to the anterior tip of the element, having a dentary with a Meckelian groove that does not extend through the symphysis, and in lacking a notch on the anterior trochanter of the femur. Kwanasaurus shares with Eucoelophysis a pronounced lateral ridge on the dentary, a dentary with a ventrally positioned Meckelian groove, a Meckelian groove that does not extend through the symphysis, broad and coarsely denticulate folidont teeth, a strongly “saddle-shaped” ilium, and the absence of a trochanteric shelf; Kwanasaurus differs from Eucoelophysis in having a more robust dentary and a fourth trochanter. Kwanasaurus shares with Diodorus a pronounced lateral ridge on the dentary, a dentary with a ventrally positioned Meckelian groove, a Meckelian groove that does not extend through the dentary symphysis, broad and coarsely denticulate folidont teeth, the absence of a trochanteric shelf, and a fourth trochanter; Kwanasaurus differs from Diodorus in only possessing canting on the anteriormost dentary teeth, possessing distinct torsion between the proximal and distal ends of the humerus, and in lacking a notch on the anterior trochanter of the femur. Kwanansaurus differs from Ignotosaurus in possessing a strongly “saddle-shaped” ilium. Kwanasaurus shares with Lutungutali a fourth trochanter on the femur; Kwanasaurus differs from Lutungutali in possessing a strongly “saddle-shaped” ilium with an elongate and flattened preacetabular process. Kwanasaurus shares with Technosaurus a dentary with a ventrally positioned Meckelian groove, and broad and coarsely denticulate folidont teeth; Kwanasaurus differs from Technosaurus in possessing a distinct lateral ridge on the dentary. Kwanasaurus shares with Soumyasaurus a dentary with a ventrally positioned Meckelian groove; Kwanasaurus differs from Soumyasaurus in posessing a much more robust dentary, a pronounced lateral ridge on the dentary, and broad and coarsely denticulate folidont teeth.

Kwanasaurus williamparkeri sp. nov.

LSID. urn:lsid:zoobank.org:act:25A4AE71-56B3-4797-B30D-1FA1D37E1F3F

Etymology. Honors friend and colleague Bill Parker, whose research has helped to greatly clarify our understanding of Late Triassic dinosauromorph diversity in the western United States.

Holotype. DMNH EPV.65879 (Figs. 8A–8H), a partial left maxilla.

Figure 8 Kwanasaurus williamparkeri maxillae.

(A) Holotype (DMNH EPV.65879) left maxilla stereopairs of lateral view, (B) interpretive drawing of same, (C) stereopairs of medial view, (D) interpretive drawing of same, (E) stereopairs of dorsal view, (F) interpretive drawing of same, (G) stereopairs of ventral view, (H) interpretive drawing of same, (I) DMNH EPV.63650, right maxilla stereopairs of lateral view, (J) interpretive drawing of same, (K) stereopairs of medial view, (L) interpretive drawing of same, (M) stereopairs of dorsal view, (N) interpretive drawing of same, (O) stereopairs of ventral view, (P) interpretive drawing of same. Hatching indicates broken bone surface, dotted lines indicate broken bone edge. Dark gray areas filled with matrix. See text for abbreviations. Scale bars = 2 cm.

Type horizon and locality. Locality DMNH 4340 (Burrow Cliff), “red siltstone member” of the Chinle Formation (Upper Triassic, Norian and/or Rhaetian), northern Colorado, USA.

Referred specimens. (see Table 1 for localities) DMNH EPV.63650 (Figs. 8I–8P), partial right maxilla; DMNH EPV.125921 (Figs. 9A–9H), partial left maxilla; DMNH EPV.125923 (Figs. 9I–9P), partial right maxilla; DMNH EPV.63136 (Fig. 10), almost complete left dentary; DMNH EPV.63135 (Figs. 11A–11D), partial right dentary; DMNH EPV.57599 (Figs. 11E–11F), partial ?right dentary; DMNH EPV.65878 (Figs. 11G–11I), partial right dentary; DMNH EPV.63660 (Figs. 11J–11L), anterior left dentary; DMNH EPV.43577 (Fig. 12A), isolated tooth; DMNH EPV.63142 (Fig. 12B), isolated tooth; DMNH EPV.63143 (Fig. 12C), isolated tooth; DMNH EPV.63843 (Fig. 12D), isolated tooth; DMNH EPV.63661 (Fig. 12E), isolated tooth; DMNH EPV.125922 (Fig. 12F), isolated tooth; DMNH EPV.59302 (Fig. 13), nearly complete left humerus; DMNH EPV.48506 (Fig. 14), complete left ilium; DMNH EPV.63653 (Figs. 15A–15C), partial left ilium; DMNH EPV.52195 (Figs. 15D–15G), partial left ilium; DMNH EPV.34579 (Fig. 16), nearly complete left femur; DMNH EPV.54828 (Figs. 17A–17E), proximal right femur; DMNH EPV.44616 (Figs. 17F–17J), proximal right femur; DMNH EPV.56651 (Figs. 17K–17O), proximal left femur; DMNH EPV.125924 (Figs. 18A–18E), proximal right femur; DMNH EPV.63874 (Figs. 18F–18J), proximal left femur; DMNH EPV.63139 (Figs. 19A–19E), proximal left femur; DMNH EPV.59311 (Figs. 19F–19J), badly worn proximal right femur; DMNH EPV.59301 (Figs. 19K–19O), proximal left femur; DMNH EPV.67956 (Fig. 20), distal left femur.

Figure 9 Kwanasaurus williamparkeri maxillae.

(A) DMNH EPV.125921, left maxilla stereopairs of lateral view, (B) interpretive drawing of same, (C) stereopairs of medial view, (D) interpretive drawing of same, (E) stereopairs of dorsal view, (F) interpretive drawing of same, (G) stereopairs of ventral view, (H) interpretive drawing of same, (I) DMNH EPV.125923, right maxilla stereopairs of lateral view, (J) interpretive drawing of same, (K) stereopairs of medial view, (L) interpretive drawing of same, (M) stereopairs of dorsal view, (N) interpretive drawing of same, (O) stereopairs of ventral view, (P) interpretive drawing of same. Hatching indicates broken bone surface or putty reconstruction, dotted lines indicate broken bone edge. Dark gray areas filled with matrix. See text for abbreviations. Scale bar = 1 cm.

Figure 10 Kwanasaurus williamparkeri DMNH EPV.63136 left dentary.

(A) Stereopairs of lateral view, (B) interpretive drawing of same, (C) stereopairs of medial view, (D) interpretive drawing of same, (E) stereopairs of dorsal view, (F) interpretive drawing of same, (G) stereopairs of ventral view, (H) interpretive drawing of same. Hatching indicates broken bone surface, dotted lines indicate broken bone edge. Dark gray areas filled with matrix. See text for abbreviations. Scale bars = 2 cm.

Figure 11 Kwanasaurus williamparkeri dentaries.

(A) DMNH 63135 right dentary stereopairs of lateral view, (B) interpretive drawing of same, (C) stereopairs of medial view, (D) interpretive drawing of same, (E) DMNH EPV.57599 right? dentary in lateral view, (F) same in medial view, (G) DMNH EPV.65878 left? dentary, lateral view, (H) same in medial view, (I) same in dorsal view, (J) DMNH EPV.63660 left dentary in lateral view, (K) same in medial view, (L) same in dorsal view. See text for abbreviations. Scale bar = 1 cm.

Figure 12 Isolated folidont teeth probably belonging to Kwanasaurus williamparkeri.

(A) DMNH EPV.43577 in (left to right) labial, lingual, edge-on, and occlusal views. (B) DMNH EPV.63142 in (left to right) labial, lingual, edge-on, and occlusal views. (C) DMNH EPV.63143 in (left to right) labial, lingual, edge-on, and occlusal views. (D) DMNH EPV.63843 in (left to right) labial, lingual, edge-on, and occlusal views. (E) DMNH EPV.63661 in (left to right) labial, edge-on, and occlusal views. (F) DMNH EPV.125922 in (left to right) labial, lingual, edge-on, and occlusal views.

Figure 13 Kwanasaurus williamparkeri left humerus (DMNH EPV.59302) stereopairs.

(A) Proximal view (anterior side facing up), (B) anterior view, (C) medial view, (D) posterior view, (E) lateral view, (F) distal view (anterior side facing up), (G) drawing of overlapping proximal and distal ends showing degree of torsion. See text for abbreviations. Scale bar = 2 cm.

Figure 14 Kwansaurus williamparkeri left ilium (DMNH EPV.48506).

(A) Stereopairs of lateral view, (B) interpretive drawing of same, (C) stereopairs of medial view, (D) interpretive drawing of same, (E) stereopairs of dorsal view, (F) interpretive drawing of same, (G) stereopairs of ventral view, (H) interpretive drawing of same. See text for abbreviations. Dotted lines indicate breaks, dashed lines outline sacral rib attachments. Scale bar = 2 cm.

Figure 15 Kwanasaurus williamparkeri ilia.

(A) DMNH EPV.63653, mostly complete left ilium in lateral view, (B) medial view, (C) ventral view, (D) DMNH EPV.52195, stereopairs of partial left ilium in lateral view, (E) medial view, (F) dorsal view, (G) ventral view. See text for abbreviations. Scale bar = 2 cm.

Figure 16 Kwanasaurus williamparkeri left femur (DMNH EPV.34579) stereopairs.

(A) Proximal view, (B) distal view, (C) anterolateral view, (D) anteromedial view, (E) posteromedial view, (F) posterolateral view. See text for abbreviations. Scale bar = 2 cm.

Figure 17 Kwanasaurus williamparkeri proximal femora, larger specimens.

(A) DMNH EPV.54828, right femur stereopairs, proximal view, (B) anterolateral view, (C) anteromedial view, (D) posteromedial view, (E) posterolateral view, (F) DMNH EPV.44616, right femur stereopairs, proximal view, (G) anterolateral view, (H) anteromedial view, (I) posteromedial view, (J) posterolateral view, (K) DMNH EPV.56651, left femur in proximal view, (L) anterolateral view, (M) anteromedial view, (N) posteromedial view, (O) posterolateral view. See text for abbreviations. Scale bar = 2 cm.

Figure 18 Kwanasaurus williamparkeri proximal femora, larger specimens.

(A) DMNH EPV.125924, right femur stereopairs in proximal view, (B) anterolateral view, (C) anteromedial view, (D) posteromedial view, (E) posterolateral view, (F) DMNH EPV.63874, left femur stereopairs in proximal view, (G) anterolateral view, (H) anteromedial view, (I) posterolateral view, (J) posterolateral view. See text for abbreviations. Scale bar = 2 cm.

Figure 19 Kwanasaurus williamparkeri proximal femora, smaller specimens.

(A) DMNH EPV.63139 left femur stereopairs in proximal view, (B) anterolateral view, (C) anteromaedial view, (D) posteromedial view, (E) posterolateral view, (F) DMNH EPV.59311 left femur in proximal view, (G) anterolateral view, (H) anteromedial view, (I) posteromedial view, (J) posterolateral view, (K) DMNH EPV.59301 left femur in proximal view, (L) anterolateral view, (M) anteromedial view, (N) posteromedial view, (O) posterolateral view. See text for abbreviations. Scale bar = 2 cm.

Figure 20 Kwanasaurus williamparkeri distal femur DMNH EPV.67956.

(A) Distal view, (B) lateral view, (C) anterior view, (D) medial view, (E) posterior view. Scale bar = 2 cm.

Diagnosis. As for genus, by monotypy.

Description and discussion. Silesaurids (non-dinosaurian dinosauriforms) are the most abundant dinosauromorphs in the Eagle Basin, although assigning elements to a particular alpha taxon is problematic for several reasons:Nearly all Eagle Basin specimens are isolated elements, reducing the number of potential autapomorphies that can be identified for any individual.

Few alpha taxon autapomorphies have been identified within Silesauridae (Peecook et al., 2013; Langer & Ferigolo, 2013; Breeden et al., 2017) with the exception of Lewisuchus (Bittencourt et al., 2014) and Asilisaurus (Nesbitt et al., 2010).

Character state polarities within Silesauridae are currently largely unresolved so that the topology of Sulcimentisauria, the sister clade to Asilisaurus, is highly variable between analyses, and taxa often fall into a polytomy (Nesbitt et al., 2010; Kammerer, Nesbitt & Shubin, 2012; Peecook et al., 2013; Sarigül, Agnolin & Chatterjee, 2018). Moreover, character state polarities are, at least in some cases, subject to both ontogeny and intraspecific variation (Piechowski, Tałanda & Dzik, 2014; Griffin & Nesbitt, 2016a, 2016b).

However, within the Eagle Basin collection, homologous elements with silesaurid apomorphies tend to share character states distinguishing these specimens from previously described silesaurid taxa. This is taken as circumstantial evidence that the Eagle Basin silesaurid material belongs to a single alpha taxon. Similar apomorphy-based logic has been applied to other silesaurid taxa where the holotype consists of a single element, and an overall picture of skeletal anatomy is cobbled together from isolated elements (Nesbitt et al., 2010; Kammerer, Nesbitt & Shubin, 2012; Langer & Ferigolo, 2013: p. 355; Peecook et al., 2017: pp. 29, 32). While far from ideal, this approach allows an at least provisional combination of phylogenetically informative character states to be assembled. These can be used to formulate phylogenetic hypotheses that are subject to potential falsification and revision by the discovery of associated material.

Maxilla

Four incomplete silesaurid maxillae are known from the Eagle Basin Chinle Formation. The holotype is DMNH EPV.65879 (Figs. 8A–8H), a left element from one of the largest individuals with a preserved anteroposterior length of 56 mm. The other three specimens are much smaller with a preserved length of 30–35 mm: right elements DMNH EPV.63650 (Figs. 8I–8P) and DMNH EPV.125921 (Figs. 9A–9H), and left element DMNH EPV.125923 (Figs. 9I–9P). All specimens can be assigned to Silesauridae due to the teeth being ankylosed into the sockets (Nesbitt et al., 2010; Langer et al., 2013), and they can all be assigned to Kwanasaurus based on their robust nature and the distinctive flange on the medial surface absent in other silesaurids (see below). The maxilla has been previously described in Lewisuchus (Bittencourt et al., 2014), Silesaurus (Dzik, 2003), Sacisaurus (Langer & Ferigolo, 2013), and Lutungutali (Peecook et al., 2017) (Figs. 21C–21I).

Figure 21 Silesaurid left maxillae.

(A) Kwanasaurus williamparkeri (composite reconstruction based on DMNH EPV.65879 and DMNH EPV.63650) in lateral view, (B) same in medial view, (C) Lewisuchus admixtus (PULR 01 redrawn from Bittencourt et al., 2014, fig. 1) in lateral view reversed, (D) same in medial view, reversed, (E) Silesaurus opolensis (ZPAL Ab III/361/26) in lateral view reversed, (F) same in medial view, reversed, (G) Sacisaurus agudoensis (MCN PV 10050) in lateral view, reversed, (H) Lutungutali sitwensis (NHCC LB649) in lateral view reversed, (I) same in medial view, reversed. Scale bar for (A–F) = 1 cm; scale bar for (G–I) = 0.5 cm. Dashed lines indicate broken edges. Arrows indicate posterior end of tooth row based on published information and figures.

All Eagle Basin elements preserve most of the tooth-bearing body of the maxilla. DMNH EPV.65879 and DMNH EPV.125921 lack the anteriormost tip of the element (Figs. 8A–8H and 9A–9F) and DMNH EPV.63650 and DMNH EPV.125923 lack the posterior tip (Figs. 8I–8P and 9I–9P). DMNH EPV.65879 and DMNH EPV.63650 preserve the base of the ascending process (asm in Figs. 8A–8F), which is completely missing in the other specimens; however, in DMNH EPV.125921 the process, although apparently lost, was reconstructed by pushing epoxy putty into the impression of the medial surface preserved in matrix (Figs. 9A–9F).

The main body and posterior process of the maxilla is a far dorsoventrally deeper, anteroventrally shorter, and more robust element than occurs in other silesaurid taxa (Figs. 8–9 and 21). In lateral view, the main tooth-bearing body of the maxilla is slightly dorsally emarginated by the antorbital fossa (see below) between about the third or fourth and sixth tooth positions (Figs. 8–9). DMNH EPV.125921 is somewhat more gracile in appearance compared to the other Eagle Basin specimens (Figs. 9A–9F), but still more robust than other silesaurids (Fig. 21). In all specimens, there is a row of small subcircular to ovate foramina on the lateral surface of the maxilla immediately above the tooth row that extends the length of the tooth-bearing segment. The foramina do not have a one to one relationship with the alveoli (Figs. 8A–8B, 8I–8J, 9A–9B and 9I–9J). In DMNH EPV.65879 and DMNH EPV.125923, additional scattered subcircular and elongate foramina of similar size occur above this lower row (Figs. 8A–8B and 9I–9J); this is not clearly evident in the other specimens.

In all specimens, the medial (lingual) surface of the main tooth-bearing body of the maxilla bears a row of larger foramina (rf in Figs. 8–9) that extend the length of the element just above the tooth sockets, and have a clear one to one relationship with the alveoli. These foramina are similar to those seen in some thyreophoran dinosaurs (Edmund, 1960; Colbert, 1981). All foramina are well-developed and smooth-walled, and might have been openings for nerve and vasculature to the alveolus instead of resorption pits, which are generally formed by the disappearance or remodeling of the tooth root and bone during the tooth replacement process. Consequently we use the term replacement foramina sensu Edmund (1960) for these openings instead of resorption pits. These foramina are particularly compressed and elongate above the first four to five tooth positions, and become more broadly ovate to circular posteriorly. In DMNH EPV.65879, the first five elongate replacement foramina lie within a clearly defined groove (in Figs. 8C–8D; largely concealed by the medial flange), which shallows and ends at the sixth replacement foramen; this groove is absent in the smaller specimens, where the foramina are also relatively large. Foramina set within a groove occur in the same position in Silesaurus (Fig. 21F; Dzik, 2003: fig. 5A), and Lutungutali (Figs. 21H–21I; Peecook et al., 2017: fig. 10C–10D). Other numerous tiny foramina are scattered across the medial surface.

The anteriormost end of the lateral surface of the maxilla is slightly inset and angled medially relative to the main body of the element above the first tooth position. This probably represents the area overlapped laterally by the premaxilla (pm.ar in Figs. 8–9). The same condition seems to be present in Silesaurus (Fig. 21E; Dzik, 2003: fig. 5B), and an anteriorly facing concavity also occurs here in Sacisaurus (Fig. 21G; Langer & Ferigolo, 2013). In Lewisuchus, the “shallow notch” (labeled “pm.ar?” in Fig. 21C) at the base of the ascending process of the maxilla (Bittencourt et al., 2014: p. 191) may be homologous to that concavity. This inset region terminates anteriorly with a short pointed prong, the anteromedial process (amp in Figs. 8–9 and 21; Prieto-Marquez & Norell, 2011), originating immediately anterior to the first tooth position, which also occurs in Silesaurus (Fig. 21E; Dzik, 2003: fig. 5A), Lewisuchus (Fig. 21C; Bittencourt et al., 2014 described this as the “maxillary cranial process”); and other archosaurs. This region is either not well-preserved in Sacisaurus, or the process is extremely short in that taxon (Fig. 21G; Langer & Ferigolo, 2013: fig. 2). The anteromedial process is best-preserved in DMNH EPV.65879 and especially DMNH EPV.125923, and has a distinctly hooked shape in dorsal view (Figs. 8E–8F and 9O–9P).

The medial surface of the anteriomedial process bears a sharp longitudinal crest (vo.ar in Figs. 8–9), probably representing the vomerine flange (Prieto-Marquez & Norell, 2011). In the three smaller specimens, the vomerine flange is very sharp, but in DMNH EPV.65879 (Figs. 8C–8D) it is thicker with longitudinal striations along its ventral surface. In DMNH EPV.65879 and DMNH EPV.125923 the process projects medially just anterior to the first tooth position (Figs. 8G–8H and 9O–9P). A thick vomerine flange is also present in Lewisuchus (Fig. 21D) and Silesaurus (Fig. 21F). Langer & Ferigolo (2013: p. 355), described (but did not figure) a “short/plate-like palatal ramus” that may also be the vomerine flange in Sacisaurus specimen MCN PV10091.

Only the very base of the ascending process of the maxilla (asm in Figs. 8–9) remains in DMNH EPV.65879 (Figs. 8A–8D) and DMNH EPV.125923 (Figs. 9P–9N), but the ascending process is slightly more complete in DMNH EPV.63650 (Figs. 8I–8L), although badly damaged, and the impression of the medial surface is preserved in DMNH EPV.125921 (Figs. 9C–9F). The ascending process is extremely thin in DMNH EPV.63650, and this seems to have been the case in the other specimens as well judging by the width of the broken edge (brk in Figs. 8F and 9N). In all specimens, the ascending process originated at least as far anteriorly as the first tooth position, rising steeply posterodorsally from the anteromedial process or just posterior to it; the anterior edge of the ascending process also seems to rise steeply as in Sacisaurus (Fig. 21G; Langer & Ferigolo, 2013) and possibly Silesaurus (Figs. 12E–12F; Dzik, 2003: fig. 6) in contrast to the more gently posterodorsally sloping ascending process of Lewisuchus (Figs. 12C–12D; Bittencourt et al., 2014: fig. 1). The ascending process in DMNH EPV.63650 is somewhat dorsomedially inclined (Figs. 8M–8N) though this is not evident in DMNH EPV.125921 (Figs. 9E–9F). The posteroventral edge of the ascending process in DMNH EPV.63650 and DMNH EPV.125923 is intact, and slopes to join the dorsal edge of the main body of the maxilla above about the sixth tooth position (Figs. 8I–8L and 9I–9L). The ascending process seems to be anteroposteriorly shorter in other silesaurids (Fig. 21).

Most specimens except for DMNH EPV.63650 preserve only a tiny remnant of the anterior edge of the antorbital fossa (afo in Figs. 8–9). However, DMNH EPV.63650 preserves what seems to be a nearly complete antorbital fossa (=the “recessed medial lamina of the dorsal process” sensu Prieto-Marquez & Norell, 2011) that embays the posterior half or so of the lateral surface of the ascending process (Figs. 8I–8J). The fossa is subtriangular with slightly convex anterior and ventral margins. The ventral margin of the antorbital fossa parallels the tooth margin as in most silesaurids other than Silesaurus (Fig. 21E) where the fossa descends to almost contact the dental margin (Peecook et al., 2017: p. 26); due to the robustness of the maxilla in Kwanasaurus, the ventral margin of the fossa is further from the dental margin than in any other silesaurid (Fig. 21A). The ventral margin extends between about the fourth and seventh tooth positions (also seen in DMNH EPV.125923; Figs. 9I–9J), while the anterior margin did not contact the nasal. In DMNH EPV.65650 a distinct swollen area occurs at the ventral margin of the fossa above the fourth tooth position (Figs. 8I–8J). In the same specimen, an irregular hole with clearly broken edges has removed most of the surface of the fossa in this specimen, so it is unclear if there was a promaxillary fenestra as in Sacisaurus (pmf in Fig. 21G; Langer & Ferigolo, 2013). The medial side of the posterior edge of the ascending process is slightly thickened by a faint ridge in DMNH EPV.65650 (Figs. 8K–8L); in both that specimen and the reconstructed DMNH EPV.125921, the anterior part of the medial surface bears a distinct sulcus (sul in Figs. 8K–8L and 9C–9D).

The most striking feature of the medial (lingual) side of the maxilla is an enormous medial flange that is fully preserved in both DMNH EPV.65879 DMNH EPV.63650 and partially preserved in the other specimens (mef in Figs. 8C–8D, 9C–9D and 9K–9L). In all specimens, the flange originates as a thick ridge that crests just posterodorsally from the vomerine flange, and in the more complete specimens (Fig. 8) descends posteroventally to become a sharper-edged, subtriangular flange that reaches its greatest breadth below the fifth and sixth tooth positions. Posterior to this, the edge of the flange ascends posterodorsally to become a smaller and even sharper-edged crest representing the palatine flange (see below). The medial flange is clearly absent in Silesaurus (Fig. 21F; Dzik, 2003: fig. 5A), Lewisuchus (Fig. 21D; Bittencourt et al., 2014), and Lutungutali (Fig. 21I; Peecook et al., 2017: fig. 10C) and the condition is unknown from other silesaurids, including Sacisaurus for which the medial surface of the only known complete maxilla (MCN PV10050) is concealed (Langer & Ferigolo, 2013). To our knowledge, nothing similar has been described in any other Triassic dinosauromorphs, where the vomer and palatine articulations are usually fully separated rather than being joined by any kind of crest (Colbert, 1989; Dzik, 2003; Prieto-Marquez & Norell, 2011). It is tempting to speculate that the medial flange in the Eagle Basin specimens is actually a separate element, perhaps the palatine fused to the maxilla, but it lacks any obvious medial articular surface for the pterygoid, and no trace of a continuous suture can be clearly discerned separating the flange from the main body of the maxilla in either specimen, even in the smaller (and likely less mature) specimens. Moreover, the probable sutural surface for the palatine can be discerned in the holotype (see below).

In DMNH EPV.65879 there is a complex series of crests, grooves, ridges, and rugosities on the dorsal and medial surfaces of the posterior ramus of the maxilla probably representing the contacts for the jugal, lacrimal, and palatine (ju.la.ar in Figs. 8C–8F). This region is far more complex in DMNH EPV.65879 than in Lewisuchus, Silesaurus (Figs. 21D and 21F), or the smaller Kwanasaurus specimens (Figs. 8K–8N and 9C–9F). This area is concealed by matrix in DMNH EPV.125923 (Figs. 9K–9N). However, the morphology of this area is remarkably similar to that of the Plateosaurus specimen described by Prieto-Marquez & Norell, (2011: figs. 4–5), and our interpretation is modeled after theirs. A prominent flange rises from the lateral side of the dorsal surface of the posterior ramus, convex on the lateral surface and concave on the medial surface; we refer to it as the posterolateral flange (plf in Figs. 8B, 8D and 21A–21B). It is tempting to suggest that this crest represents part of the jugal or lacrimal, but it seems to clearly be part of the maxilla with no trace of a suture. In lateral view, this flange would have partly concealed the anterior end of the articulated jugal in lateral view. No similar flange occurs in the smaller Eagle Basin specimens (Figs. 8I–8P and 9), so it is possible that this is a feature that develops with maturity.

In DMNH EPV.65879, two deep, longitudinal, dorsomedially-facing grooves separated by a ridge occur on the dorsal surface of the posterior end of the maxilla, above the posterior termination of the medial flange (ju.la.ar in Figs. 8C–8F). These medial and lateral grooves probably represent the jugal and lacrimal articulations, respectively. Both originate above the 9th tooth position, but the lateral groove extends to the posterior end of the maxilla, while the medial groove only extends as far as the 11th tooth position. Ventral to the medial (lacrimal?) groove, the medial surface of the posterior process is covered with pits and striations that may also be part of the lacrimal articulation. The lateral surface of the posterior tip of the maxilla bears small tuberosities (Figs. 8A–8B) suggesting a tight sutural contact with the jugal.

In DMNH EPV.65879 there is a distinct triangular embayment occurring slightly more anteriorly along the edge of the medial flange but just posterior to the apex of the flange (pa.ar in Figs. 8C–8D). This region probably represents the articulation with the palatine, in which case the palatine had a very broad contact with posterior edge of the medial flange of the maxilla. This sutural surface is not evident in any of the smaller specimens, although in DMNH EPV.123923 the region is not fully prepared.

In DMNH EPV.65879, the main tooth-bearing body of the maxilla seems to have a completely preserved tooth row with 12 tooth positions, with fully emergent teeth in the 1st, 2nd, and 4th alveoli (Figs. 8A–8D and 8G–8H). This is similar to the maxillary tooth counts in Silesaurus (11; Dzik, 2003) and Sacisaurus (10; Ferigolo & Langer, 2007) but considerably less than in Lewisuchus (20; Bittencourt et al., 2014). The main body of the maxilla is missing past the ninth tooth position in DMNH EPV.63650 and not well-preserved in the other two specimens, but all seem to have had minimally nine teeth and probably more. The posteriormost alveoli in the maxilla are indicated by an arrow in Fig. 21; the alveoli extend almost to the posterior end of the posterior ramus of the maxilla in Kwanasaurus (Figs. 21A–21B); this is also the case in Lewisuchus (Figs. 21C–21D; Bittencourt et al., 2014: fig. 1), but not in Silesaurus or Sacisaurus, where the posteriormost part of the maxilla seems to be edentulous (Figs. 21E–21G; Dzik, 2003: fig. 6; Langer & Ferigolo, 2013).

In DMNH EPV.63650 and DMNH EPV.125923 there is a deep depression above the anteriormost teeth that contains a series of smaller subcircular depressions (rp in Fig. 9N; not visible in Figs. 8M–8N due to the ascending process being preserved). In DMNH EPV.65879 this same region is contains a thickened area with circular areas of spongy bone occurring over the 2nd and 3rd tooth positions, and a poorly preserved pit seems to occur above the 1st tooth position (Figs. 8C–8F). These depressed areas seem to be associated with the dorsal ends of the tooth roots; indeed, in DMNH EPV.125923 the root of the emerging third tooth crown projects from the dorsal surface of the medial flange (rt in Figs. 9I–9N). These depressions and areas of spongy bone might be resorption pits. Possible resorption pits are also evident on the medial side of the ascending process in Lewisuchus (Fig. 21D), Silesaurus (Fig. 21F), and Lutungutali (Fig. 21I). The pattern of tooth replacement will be discussed in more detail below. In all specimens, the ventral side of the medial flange also defines an elongate depression with a series of deeper subcircular depressions occurring beneath the broadest part of the flange (best seen in Figs. 8G–8H below where “mef” is labeled), which do not have a one to one relationship with the tooth positions.

The dorsal surface of the main body of the maxilla in DMNH EPV.65879 is covered with deep pits and grooves of uncertain nature (the dark patches near the region marked “brk” in Fig. 8F). Just anterior to the two grooves representing the jugal and lacrimal articulation is another deep groove, the posterior part of which seems to be surrounded by finished bone (fo in Fig. 8F), but the anterior part and pits appear to be broken bone, and occur where the antorbital fossa of the ascending process occurs in DMNH EPV.63650 and DMNH EPV.125923. It is therefore suggested that these represent an originally closed canal and/or cavities that were covered by the ascending process or exited its base as a foramen. A similarly positioned foramen seems to occur on the dorsal surface of the maxilla in Silesaurus (Fig. 21F; Dzik, 2003: fig. 5), but cannot be clearly discerned in other Eagle Basin specimens.

Dentary and angular

Two nearly complete silesaurid dentaries are known from the Eagle Basin; DMNH EPV.63136 (a left; Fig. 10) and DMNH EPV.63135 (a right; Figs. 11A–11D). DMNH EPV.63136 is the most complete dentary described for a silesaurid, as it seems to completely preserve both the anteriormost and posteriormost ends, unlike all other described silesaurid dentaries (Fig. 22; Irmis et al., 2007a; Nesbitt, Irmis & Parker, 2007; Nesbitt et al., 2010; Kammerer, Nesbitt & Shubin, 2012; Langer & Ferigolo, 2013). DMNH EPV.63136 has a preserved anteroposterior length of 36 mm, and a maximum preserved dorsoventral height (not counting the tooth crowns) of 11 mm. DMNH EPV.63135 is missing an uncertain amount of the anterior and posterior ends, but based on comparison with the more complete specimen, the most anteriorly preserved tooth crown is probably in the third tooth position; the specimen has a preserved anteroposterior length of 34 mm, and a maximum preserved dorsoventral height of eight mm. Two other dentaries, DMNH EPV.57599 (a possible right; Figs. 11E–11F), and DMNH EPV.65878 (a possible left; Figs. 11G–11I), are missing an uncertain amount of the anterior and posterior ends, while DMNH EPV.63660 is a left anterior end (Figs. 11J–11L). All of these specimens seem to represent individuals of comparable size or smaller than the more complete dentaries.

Figure 22 Silesaurid left dentaries.

(A) Kwanasaurus williamparkeri (based primarily on DMNH EPV.63136) in lateral view, (B) same in medial view, (C) Asilisaurus kongwe (NMT R89) in lateral view, (D) same in medial view, (E) Eucoelophysis baldwini (GR 224) in lateral view, (F) same in medial view, (G) Technosaurus smalli (TTU P-9021, reversed) in lateral view, (H) same in medial view (also reversed), (I) Sacisaurus agudoensis (composite based on MCN PV10042 and MCN PV10043) in lateral view, (J) same in medial view, (K) Silesaurus opolensis (ZPAL AbIII/361/26) in lateral view, (L) same in medial view, (M) Diodorus scytobrachion (MNHM-ARG 30) in lateral view (reversed), (N) same in medial view (also reversed), (O) Soumyasaurus aenigmaticus (TTU-P1125b) in lateral view, (P) same in medial view. Dashed lines indicate broken edges. Unshaded regions indicate the surface of the specimen is not exposed. All scale bars = 1 cm.

As with the maxillae, all specimens can be assigned to Silesauridae due to the teeth being ankylosed into the sockets (Nesbitt et al., 2010; Langer et al., 2013). These dentaries can also be assigned to Sulcimentisauria, the clade containing all known silesaurids exclusive of Asilisaurus based on the following apomorphies: Meckelian groove lies near the ventral margin of the dentary (Mk in Figs. 10–11), and dentary teeth have constrictions below the crown (Appendix 3; Nesbitt et al., 2010). Moreover, in DMNH EPV.63135 and DMNH EPV.63136 the dorsal edge of the dentary is clearly concave rather than convex, and the dentary teeth crowns are short and sub-triangular with large denticles (Figs. 10 and 11A–11D) rather than recurved or peg-like, which also distinguishes these taxa from Asilisaurus (Nesbitt et al., 2010), Silesaurus (Figs. 22K–22L; Dzik, 2003), and Soumyasaurus (Figs. 22O–22P; Sarigül, Agnolin & Chatterjee, 2018). In DMNH EPV.63136 and DMNH EPV.63660, the only specimens to preserve the very tip of the dentary, the anterior tip is a sharp, edentulous point (Figs. 10 and 11J–11L), another silesaurid feature (Nesbitt et al., 2010).

The dentary of Kwanasaurus seems to be distinctly deeper than the relatively slender dentaries of Eucoelophysis (Figs. 22E–22F), Sacisaurus (Figs. 22I–22J), and Soumyasaurus (Figs. 22O–22P). The ventral margins of DMNH EPV.63135, DMNH EPV.63136, and DMNH EPV.63660 are slightly convex (Figs. 10A–10D, 11A–11D and 11J–11L); the other specimens are too incomplete to be certain if they share this feature. Viewed dorsally or ventrally, the two most complete dentaries also curve slightly posterolaterally, suggesting that this shape is natural; DMNH EPV.63136 is constricted at the edentulous tip and symphysis, with the rest of the mandible flaring posterolaterally (Figs. 10E–10H).

The lateral surface of all the dentaries except DMNH EPV.63660 (which only possesses the anterior tip) bears a distinct lateral ridge roughly midway between the dorsal and ventral margins (lr in Figs. 10–11 and 22). In DMNH EPV.63136 the ridge originates approximately under the fourth alveolus, and terminates posteriorly at the anterior end of the mandibular fenestra, roughly below the 9th and 10th tooth positions (Figs. 10A–10B). In DMNH EPV.63135 the ridge originates beneath the second preserved alveolus and is most prominent under the eighth tooth position (Figs. 11A–11B). Among other silesaurids, a distinct lateral ridge is reported only for Diodorus (Fig. 22M; Kammerer, Nesbitt & Shubin, 2012), but also occurs in Eucoelophysis material from the Hayden Quarry (Fig. 22E; J. Martz, 2017, personal observation of GR 224).

A posteriorly facing foramen on the upper surface of the ridge occurs below the 9th tooth position in both DMNH EPV.63136, and DMNH EPV.63135 (fo in Figs. 10A–10B and 11A–11B). A similar posteriorly opening foramen is also known in aetosaurs (Small, 2002), and seems to also be present in Diodorus (Fig. 22M; Kammerer, Nesbitt & Shubin, 2012: fig. 1A). In DMNH EPV.57599 and DMNH EPV.63135, a canal conducted within the ridge was observed at the edges of the break in the element (in the latter specimen, it is no longer visible as the two halves of the dentary are glued together); the canal may connect to the posterior facing foramen. This canal also occurs within the ridge in Eucoelophysis (JW Martz, personal observation of GR 224). Smaller nutrient foramina exit from the dorsal surface of the ridge in both of the more complete dentaries (Figs. 10A–10B and 11A–11B) as in Diodorus (Fig. 22M; Kammerer, Nesbitt & Shubin, 2012) and Eucoelophysis (Fig. 22E; JW Martz, personal observation of GR 224); in DMNH EPV.63136 and DMNH EPV.63135 even smaller foramina exit from the ventral side of the ridge and the underside of the edentulous tip.

In DMNH EPV.63136 and DMNH EP.63660, the anterior edentulous tip of the dentary bears a distinct groove on the lateral surface that extends from the tip of the element to enter the element beneath the second tooth position (gr in Figs. 10A–10B, 11J and 22). A similar groove occurs in Silesaurus and Sacisaurus (Figs. 22I and 22K) that Dzik (2003) describes it as a “vascular canal,” and Langer & Ferigolo (2013) indicate that it originates in a “mental foramen” at the posterior end of the groove, although this is difficult to evaluate in the Kwanasaurus specimens because matrix has not been fully removed from the groove. In Sacisaurus, the groove differs from that of Silesaurus and Kwanasaurus in that it rises to the dorsal margin of the dentary (Fig. 22I; Langer & Ferigolo, 2013) rather than extending longitudinally to the tip (Figs. 22A and 22K).

The symphysis of Kwanasaurus is fully preserved in DMNH EPV.63136 and DMNH EPV.63660, and partly preserved in DMNH EPV.63135 (sy in Figs. 10C–10D, 11C–11D, 11J–11L and 22). The symphyseal surface seems to extend from below the third tooth to the anterior tip of the dentary. It is a slightly rugose “type II symphyseal surface,” as is seen in other silesaurids (Holliday & Nesbitt, 2013). There are with two distinct, non-continuous grooves occurring where the complete symphysis is preserved, one beneath the first through third tooth positions, and another emerging from an anteriorly facing foramen beneath the first tooth and extending to the anterior tip (Figs. 10C–10D and 11K). However, these grooves are not continuous with the Meckelian groove, which seems to terminate behind the dentary symphysis, below the third tooth. The Meckelian groove also terminates behind the symphysis in Diodorus (Fig. 22N; Kammerer, Nesbitt & Shubin, 2012), but allegedly extends through the symphysis in Sacisaurus and Silesaurus (Figs. 22J–22L; Dzik, 2003; Ferigolo & Langer, 2007). In DMNH EPV.63136, there is another thin groove on the edentulous tip above the anterior groove (Figs. 10C–10D).

A total of 14 tooth positions are present in the dentary of DMNH EPV.63136 (Fig. 10), seven of which contain fully erupted teeth (in positions 1, 3, 4, 6, 9, 11, and 12). This seems to represent the entire tooth row, and falls within the general range of tooth counts seen in Silesaurus (12; Dzik, 2003), Sacisaurus (15; Ferigolo & Langer, 2007), and Soumyasaurus (at least 15; Sarigül, Agnolin & Chatterjee, 2018). At least 11 tooth positions are present in the less complete DMNH EPV.63135 (Figs. 11A–11D), for which numbering of tooth position is inferred by comparison with DMNH EPV.63136.

Replacement foramina identical to the replacement foramina of the maxillae occur beneath each alveolus (rf in Figs. 10D, 11D, 11F, 11H and 22). These replacement foramina are connected by a shallow groove as far back as the 8th alveolus, with the medial surface of the dentary between the groove and the teeth being slightly inset from the rest of the medial surface; this is also seen in Silesaurus (Fig. 22L; Dzik, 2003: fig. 5E), Eucoelophysis (Fig. 22F; JW Martz, personal observation of GR 224), and Technosaurus (Fig. 22H; Martz et al., 2013: fig. 14G). The inset and groove shallow to merge with the rest of the medial surface beneath about the ninth alveolus in DMNH EPV.63136 and DMNH EPV.63135. In DMNH EPV.63135 and DMNH EPV.63878, the foramina beneath emergent crowns are elongate ovals, while pits under empty alveoli and crowns that are not fully emerged are larger and more circular (Figs. 11C–11D and 11I). This difference in shape between foramina under fully erupted and unerupted crowns is not evident in DMNH EPV.63136, where the replacement foramina generally become larger posteriorly rather than beneath empty alveoli (Figs. 10C–10D); this is the pattern also seen in DMNH EPV.65879 the holotype maxilla of Kwanasaurus (Figs. 8C–8D).

In all specimens of Kwanasaurus, the dorsal margin of the dentary is strongly depressed above empty alveoli, and raised where it is fused to emergent crowns as a striated region below the crown (Figs. 10A–10D and 11A–11D). The depression of the alveolar margin is evident in other silesaurids, especially Diodorus (Figs. 22M–22N) and Sacisaurus (Figs. 22I–22J), but more difficult to evaluate in Silesaurus (Figs. 22K–22L), Technosaurus (Figs. 22G–22H), and Eucoelophysis (Figs. 22E–22F), where the teeth are more tightly packed and/or regions without teeth are damaged.

On the medial surface of all dentaries of Kwanasaurus, the Meckelian groove extends along the ventral edge (Mk in Figs. 10C–10D, 11C–11D, 11F, 11H, 11K and 22), as in Silesaurus, Sacisaurus, Diodorus. Eucoelophysis, and Technosaurus (Figs. 22E–22N; Dzik, 2003: fig. 5E; Ferigolo & Langer, 2007: fig. 7I; Irmis et al., 2007a: fig. 2L; Kammerer, Nesbitt & Shubin, 2012; Martz et al., 2013: fig. 14G). This is not the case in Asilisaurus, where the groove is midway between the dorsal and ventral margins (Figs. 22C–22D; Nesbitt et al., 2010). In the most complete dentaries, the Meckelian groove is dorsoventrally widest posteriorly, near the mandibular fenestra (Figs. 10C–10D; 11C–11D). In DMNH EPV.63136, the groove has a maximum height of six mm high, or about 55% of the height of the dentary exclusive of the teeth and the groove narrows to almost nothing beneath the third tooth position.

Unlike any other described silesaurid dentary, in which the posteriormost part of the dentary is usually damaged or missing (Figs. 22C–22N; Dzik, 2003; Ferigolo & Langer, 2007; Nesbitt et al., 2010; Kammerer, Nesbitt & Shubin, 2012; Martz et al., 2013), DMNH EPV.63136 preserves a very thin and fragile posteroventral process forming the ventral border of the mandibular fenestra (maf in Figs. 10A–10D, 13A–13B and 22A–22B); the medial side of this process is concave and formed the lateral border of the posterior part of the Meckelian groove (Figs. 10C–10D).

In DMNH EPV.63136, the sharply pointed anteriormost tip of the angular (as in Fig. 10A–10D) is preserved in contact with the posterior end of the posteroventral process of the dentary. The posteroventral process tapers posteriorly to a sharp point that overlies the anterior tip of the angular; comparing the lateral and medial shapes of the contact between the elements suggests that the process of the dentary slightly overlapped the tip of the angular laterally (Figs. 22A–22B).

The posterior end of the tooth-bearing section of the dentary, which forms the anterodorsal border of the mandibular fenestra, is also better preserved in DMNH EPV.63136 and DMNH EPV.63135 than in any previously described silesaurid specimen (Figs. 10, 11A–11D and 22A–22B). The dorsal surface of this process is a sharp edge behind the final (13th) dentary tooth. The ventral surface of the process is embayed by a deep groove. A distinct notch occurs on the lateral surface of the process below or just behind the 13th tooth position that probably received the anterior tip of the surangular (sa.ar in Figs. 10A–10B, 10E–10H and 22). The posterodorsal process seems to be somewhat deeper relative to the rest of the dentary in Sacisaurus specimen MCN PV10043 compared to Kwanasaurus (Figs. 22I–22J; Langer & Ferigolo, 2013: fig. 4A).

Tooth morphology

In addition to the emergent tooth crowns in the maxillae and dentaries just described (Figs. 8–11), there are six isolated teeth with the same crown morphology: DMNH EPV.43577 (Fig. 12A), DMNH EPV.63142 (Fig. 12B), DMNH EPV.63661 (Fig. 12C), DMNH EPV.63143 (Fig. 12D), DMNH EPV.63843 (Fig. 12E), and DMNH EPV.125922 (Fig. 12F). The referral of the isolated teeth to Silesauridae must be considered extremely tentative, based on their resemblance to those in the maxillae and dentaries of Kwanasaurus rather than the presence of unique silesaurid dental synapomorphies.

Nearly all maxillary, dentary, and isolated crowns are lanceolate, somewhat labially-lingually constricted (more at the tip than near the base) with a faint midline ridge and swollen base on both surfaces that is more prominent on the lingual side (the “cingulum” of Langer & Ferigolo, 2013; but see Irmis et al., 2007b). The midline ridge bears a longitudinal groove in DMNH EPV.125922 (Fig. 12F, two leftmost images). Faint longitudinal striations occur on the lingual side of the crown in DMNH EPV.63143 (Fig. 12D, second from left), but are absent on the labial side (Fig. 12D, far left), and no striations can be discerned in other specimens; longitudinal striations are common on the crowns of other silesaurids (Dzik, 2003; Nesbitt, Irmis & Parker, 2007; Nesbitt et al., 2010). The crowns are usually asymmetrical in lingual or labial view, with the mesial (posterior) side of the base being more ventrally positioned, but not recurved. The distal (anterior) carinae are often (but not always) slightly more convex than the mesial carniae. The carinae possess coarse denticles at an acute angle to the mesial and distal edges.

Similar “phyllodont” or “folidont” (Hendrickx, Mateus & Araújo, 2015) tooth crown morphology occurs in a variety of extinct diapsids that are herbivorous or interpreted as herbivorous (Sues, 2000). Folidont tooth crowns are expanded beyond the root and lanceolate rather than recurved (Hendrickx, Mateus & Araújo, 2015). Folidont teeth also frequently possess a midline ridge extending from the base to the apex on the lingual and labial surfaces, and large denticles projecting at an angle to the tooth margin. In addition to Kwanasaurus, folidont teeth occur in Sacisaurus and Eucoelophysis (Figs. 22E–22F; Irmis et al., 2007a: fig. 2L; Langer & Ferigolo, 2013) but distinct from the non-folidont condition in Asilisaurus, Silesaurus, and Soumyasaurus in which the crowns are more conical with smaller and less distinct denticles (“conidont” sensu Hendrickx, Mateus & Araújo, 2015) (Figs. 22K–22L; Dzik, 2003, Nesbitt et al., 2010; Sarigül, Agnolin & Chatterjee, 2018). The condition is harder to assess in the holotypes of Technosaurus (TTU P-9021) and Diodorus (MNHM-ARG 30). In Technosaurus, the crowns are damaged, making the presence of denticles or “accessory cusps” (Hunt & Lucas, 1994) difficult to evaluate, but the overall crown shape is similar to Kwanasaurus and the teeth are probably folidont (Figs. 22G–22H). In Diodorus the crowns also seem to be damaged and their form is therefore difficult to assess (Figs. 22M–22N; Kammerer, Nesbitt & Shubin, 2012: fig. 1). Folidont teeth also occur in early ornithischians, early sauropodomorphs, some theropods (Barrett, 2000; Araújo, Castanhinha & Mateus, 2011; Hendrickx, Mateus & Araújo, 2015) and various enigmatic Late Triassic taxa that had been previously considered to be ornithischians (Heckert, 2002; Parker et al., 2005; Irmis et al., 2007b; Nesbitt, Irmis & Parker, 2007).

Compared to the dentary teeth, the maxillary crowns of Kwanasaurus are relatively squat and robust-looking, and the anteriormost teeth in DMNH EP.65879 and DMNH EPV.125923 are more labially-lingually swollen so that they are almost circular rather than ovate in occlusal view (Figs. 8G–8H and 9O–9P), consistent with the overall robust form of the maxillae. Denticles cannot be discerned on the crowns of DMNH EPV.63650 or DMNH EPV.125923 (Figs. 8I–8L, 8O–8P, 9I–9L and 9O–9P). In comparison, the crowns of the teeth in dentaries DMNH EPV.63135, DMNH EPV.63660, DMNH EPV.65878 (Figs. 11I and 11L), and isolated teeth DMNH EPV.43577, DMNH EPV.63843, and DMNH EPV.125922 (Figs. 12A and 12D–12F) are less swollen at the base and are more mesially-distally compressed, and are also relatively symmetrical in mesial, distal, and occlusal views.

In maxillae DMNH EPV.65879, DMNH EPV.63650, the crowns and empty alveoli become gradually smaller posteriorly (Fig. 8), indicting a posterior reduction in maxillary tooth size as in known silesaurid maxillae (Figs. 21C–21G) for Lewisuchus (Bittencourt et al., 2014), Silesaurus (Dzik, 2003: fig. 5C), and Sacisaurus (Langer & Ferigolo, 2013). This is less certain in DMNH EPV.125923 and DMNH EPV.125921, where the posterior part of the tooth row is less well-preserved (Fig. 9). There is no clear canting or recurvature in maxillary teeth.

In contrast, in the most complete dentaries of Kwanasaurus (DMNH EPV.63136 and DMNH EPV.63135) the teeth clearly increase in the size into the middle of the jaw then decrease in the posteriormost alveoli (Figs. 10 and 11A–11D) as also occurs in all known silesaurid dentaries that are sufficiently complete to evaluate (Fig. 22), specifically Diodorus, Silesaurus, Sacisaurus, and Technosaurus (Dzik, 2003: figs. 5E–5F; Kammerer, Nesbitt & Shubin, 2012; Langer & Ferigolo, 2013). In DMNH EPV.63660, the first tooth is slightly more conical than the following teeth, is slightly anteriorly canted and has a concave mesial edge making it slightly recurved (Figs. 11J–11L). The first tooth of DMNH EPV.63136 is damaged, but the third tooth is also anteriorly canted (but not recurved) due to the mesial edge being longer than the distal edge (Figs. 10A–10D). Canting also occurs in the anterior dentary teeth of Sacisaurus (Figs. 22I–22J; MCN PV10050; Langer & Ferigolo, 2013: fig. 2), and in all three preserved teeth of the holotype of Diodorus (MNHM-ARG 30; Figs. 22M–22N). The anteriormost dentary teeth are not known in Eucoelophysis or Technosaurus, so it is not known if they shared the condition, or if canting carries a significant phylogenetic signal.

None of the maxillary teeth of Kwanasaurus are sufficiently well-preserved to determine if denticle count changes with crown size, but in the dentaries and isolated crowns, larger crowns have more denticles; in dentary teeth, this means that there is a general anterior to posterior increase in denticle counts (Table S1). This relationship between crown size and denticle count also occurs in the isolated crowns. There appear to be at least four or five denticles (not all are preserved) along both the mesial and distal edges of DMNH EPV.63142, DMNH EPV.43577 and DMNH EPV.63661, but seven on each edge of DMNH EPV.63143, the largest of the isolated crowns (Fig. 12C).

The isolated crowns all preserve a single root, which appears to be nearly complete in all four specimens (Fig. 22). The relatively complete roots of DMNH EPV.63142, DMNH EPV.63661, and DMNH EPV.63143 are about twice the length of the crown. The roots taper away from the crown; they are thicker and subcircular or oval closer to the crown, where they are slightly constricted labially-lingually, and narrow to a thinner subcircular tip. In mesial and distal views the root curves slightly lingually in DMNH EPV.63136 and DMNH EPV.63135.

Tooth replacement patterns

Ankylosis of fully erupted socketed teeth to jaw (“ankylosed thecodont” or “ankylothecodont” sensu Edmund, 1969: p. 129 and Chatterjee, 1974: p. 230) occurs in the Eagle Basin specimens as in all silesaurids where tooth-bearing elements are preserved (Dzik, 2003; Nesbitt et al., 2010; Irmis et al., 2007a; Kammerer, Nesbitt & Shubin, 2012; Langer & Ferigolo, 2013; Martz et al., 2013; Peecook et al., 2017), and is a synapomorphy of Silesauridae (Nesbitt et al., 2010). Tooth replacement in the Eagle Basin material occurs in a generally alternating sequence (Figs. 8–11, Table S1; Zahnreinhen waves of replacement sensu Woerdeman, 1921), but there are complications to this pattern, as will be discussed below.

Tooth replacement occurred on the lingual side of the fully erupted crown, as is typical of amniotes (Edmund, 1969); in the fourth tooth position of the largest maxilla DMNH EPV.65879 (Figs. 8C–8D), the incoming replacement crown lies in an embayment on the lingual side of the fully emergent crown, indicating that dissolution of the medial side of the root accompanied the emergence of the replacement crown within the same socket (“iguanid” tooth replacement sensu Edmund, 1960: p. 61–62). The dorsal surface of the maxilla is damaged above the fourth tooth position, so it is not clear if the root of the replacement tooth was still intact; details of tooth replacement are more difficult to determine with thecodont dentitions as the roots are at least partially concealed, unlike pleurodont dentitions where the lingual surfaces of the roots are exposed, showing the earlier stages of root resorption (Edmund, 1969: p. 136). However, in DMNH EPV.125923 (Figs. 9I–9P), the incoming replacement tooth still possesses a root projecting above the main body of the maxilla, and the prior crown is already gone. This suggests that maxillary tooth replacement occurred as follows:The replacement tooth forms with the root projecting above the main body of the maxilla. As the tooth moves into position, the lingual side of the previously emplaced crown and the cement holding it to the alveolar margin is dissolved (as seen in DMNH EPV.65879).

The previously emplaced crown and whatever remains of the root is released while the replacement crown moves into position, the root still attached (as seen in DMNH EPV.125923).

With the replacement crown fully emplaced, at least the part of the root projecting above the main body of the maxilla is dissolved, leaving a spongy resorption pit (best seen in DMNH EPV.65879), while the tooth is ankylosed into the jaw below the crown.

It is not clear if this pattern was identical in the dentary teeth; only DMNH EPV.63135 display incoming replacement teeth (simultaneously in tooth positions 7 and 9), and the roots, if present, are concealed inside the dentary (Figs. 11A–11D). It can at least be said that they do not project below the Meckelian groove.

Tooth replacement patterns can be complex (Edmund, 1960; Whitlock & Richman, 2013), and the number and pattern of emplaced teeth shows an interesting degree of variation among silesaurids. In Kwanasaurus, there is a clear alternating pattern of tooth replacement in both the maxilla and dentary in which there are no more than two adjacent fully erupted and ankylosed crowns (Figs. 8–11), DMNH EPV.63135 shows replacement teeth coming in simultaneously on either side of a fully emergent crown (Figs. 11A–11D). An alternating pattern of replacement in which there are no more than two adjacent fully erupted crowns also occurs in some dentaries of Sacisaurus (Figs. 22I–22J; Langer & Ferigolo, 2013: figs. 3–4), and apparently the less complete holotype dentaries of Diodorus (Figs. 22M–22N; Kammerer, Nesbitt & Shubin, 2012) and Asilisaurus (Nesbitt et al., 2010: fig. 1B).

However, in another dentary of Sacisaurus (MCN PV10048; Langer & Ferigolo, 2013: fig. 5) there are three adjacent fully erupted crowns, and a maxilla assigned to that taxon has five sequential fully erupted crowns (Fig. 21G; Langer & Ferigolo, 2013: fig. 5). Silesaurus maxilla ZPAL Ab III/361/26 has four sequential fully erupted crowns, while dentary ZPAL Ab III/437/1 has five (Figs. 22K–22L; Dzik, 2003: figs. 5A–5B and 5E–5F). In the holotype dentary of Technosaurus (TTU P-11282) there are six sequential fully erupted crowns (Figs. 22G–22H; Chatterjee, 1984; Martz et al., 2013: fig. 14G).

In summary, there are silesaurid tooth-bearing elements with rows of almost entirely fully emergent teeth, others in which replacement has left blocks of three or more sequential teeth, and some in which fully emplaced crowns mostly alternate between odd and even tooth positions. It is not clear if these patterns of variation are taxonomically significant, or if different silesaurid specimens merely show the same patterns of tooth replacement at different stages; the latter seems likely given that the pattern of tooth replacement varies within Sacisaurus (Langer & Ferigolo, 2013).

Humerus

DMNH EPV.59302, a nearly complete left humerus (Fig. 13; measurements in Table S2), is remarkably similar to those of Silesaurus and Diodorus in being long, straight, very sender, and simple in form (Dzik, 2003: fig. 9B; Kammerer, Nesbitt & Shubin, 2012: fig. 2). The proximal end is not fully preserved, but the articular surface is not distinctly thickened (Fig. 13A), or as straight in anterior and posterior views (Figs. 15B and 15D) as in Silesaurus and Diodorus (Dzik, 2003: fig. 9B; Kammerer, Nesbitt & Shubin, 2012: figs. 2A1 and 2A3). In anterior and posterior views, the proximal end is only slightly expanded medially, whereas the lateral side bearing the deltopectoral crest is straight (dc in Figs. 13B and 13D). The deltopectoral crest is incompletely preserved, but seems to have been weakly developed, curved anteriorly, and did not extend distally more than 1/3rd of the length of the shaft (Fig. 13B), similar to Silesaurus and Diodorus (Kammerer, Nesbitt & Shubin, 2012) in contrast with the more distally elongate deltopectoral crests of dinosaurs (Langer & Benton, 2006). The anterior face of the proximal end is slightly concave, narrowing distally to a groove that shallows before the midpoint of the humerus (gr in Fig. 13B).

The midshaft is almost circular in cross section. The distal end is twisted so that the long axis is almost perpendicular to that of the proximal end (Fig. 13G); torsion also seems to occur to some extent in Silesaurus (Dzik, 2003: fig. 9B) but not in Diodorus, where the long axis of the proximal and distal ends are parallel (Kammerer, Nesbitt & Shubin, 2012: p. 279). The distal end is even less expanded relative to the shaft than the proximal end (Figs. 13E–13G), with no trace of entepicondylar or ectepicondylar flanges or grooves as is typical for ornithodirans (Nesbitt, 2011: character 234). Between the distal condyles (ect and ent in Fig. 13), both the anterolaterally and posteromedially facing surfaces of the distal end are concave, with the concavity extending somewhat proximally up the shaft. The concavity on the anterolateral surface is deeper, with a deep groove (gr in Fig. 13E); a similar groove also occurs in Diodorus (Kammerer, Nesbitt & Shubin, 2012: fig. 2A3).

As no unique humeri synapomorphies have been identified for Silesauridae, referral to the clade is likely but tentative and based on the strong resemblance of the element to that of Silesaurus (Dzik, 2003: fig. 9B) and Diodorus (Kammerer, Nesbitt & Shubin, 2012: fig. 2). Shuvosaurids also have extremely similar long and slender humeri with weakly developed deltopectoral crests (Long & Murry, 1995: p. 160, fig. 164; Nesbitt, 2011: character 236), but Effigia (Nesbitt, 2007: p. 45, fig. 37) and Shuvosaurus (TTU P-9001; JW Martz, personal observation; Long & Murry, 1995: fig. 164B) have large bulbous tubers on the posterior side of the proximal end that are lacking in silesaurids.

Ilium

DMNH EPV.48506, a left ilium (Fig. 14; measurements in Table S2), bears a combination of characters that suggest that it probably belongs to a silesaurid, although with some differences in relation to previously described taxa (Dzik, 2003; Nesbitt et al., 2010). DMNH EPV.63650, a slightly less complete ilium missing most of the iliac blade and end of the postacetabular process (Figs. 15A–15C) is nearly identical in size and shape. DMNH EPV.52195, a partial iliac blade with the postacetabular process preserved (Figs. 15D–15G) shares key similarities with DMNH EPV.48506, and is probably also silesaurid.

In all specimens, the iliac blade (ilb in Figs. 14A–14F, 15A–15C and 15F) is thin and strongly inclined so that it slopes ventrolaterally. This unusual orientation of the iliac blade gives the ilium a saddle-like appearance in lateral view similar to Silesaurus (Dzik, 2003), Eucoelophysis (Irmis et al., 2007a: fig. 2M), and Ignotosaurus (Martinez et al., 2012). The region is not preserved in Sacisaurus (MCN PV 10100; Langer & Ferigolo, 2013: fig. 10). The Middle Triassic silesaurids Asilisaurus and Lutungutali, which are basal to most members of Sulcimentisauria (Peecook et al., 2013; see below) differ from Kwanasaurus, Silesaurus, Eucoelophysis, and Ignotosaurus in having a taller, more vertically oriented iliac blade (Peecook et al., 2013: figs. 2–3 and 6) more like what is seen in other archosauriforms (Nesbitt, 2011: fig. 34), suggesting that this is the plesiomorphic condition for Silesauridae. The purpose of the derived condition is unclear, but it may indicate a shift in the origin of the second and third heads of the M. iliotibialis, which generally attach along the dorsal edge of the iliac crest (Carrano & Hutchinson, 2002).

In lateral view, the dorsoventrally compressed preacetabular process of both DMNH EPV.48506 and DMNH EPV.63653 is elongate and anterodorsally oriented (pra in Figs. 14A–14D and 15A–15B) as in Silesaurus (Peecook et al., 2013: fig. 6F) Eucoelophysis (Irmis et al., 2007a: fig. 2M; J.W. Martz, personal obervation of GR 225), and Ignotosaurus (Martinez et al., 2012) in contrast to the extremely thick and blunt preacetabular process in Lutungutali (Peecook et al., 2013); the process is not known for Asilisaurus or Sacisaurus. In DMNH EPV.48506, the anterior tip of the preacetabular process tapers medially to a point in dorsal view (Figs. 14E–14F). Just posterior to the tapering tip, the lateral edge of the preacetabular process in both DMNH specimens is a sharp and grooved crest in the same position as the “tuberosity” in Silesaurus and Ignotosaurus (Dzik, 2003; Martinez et al., 2012). This sharp crest loses its sharp edge and thickens to merge with the lateral surface of the ilium without quite contacting the supracetabular crest (Figs. 14A–14B and 15A).

The preacetabular process in DMNH EPV.48506 is so elongate that it extends anterior to the acetabulum (Fig. 14) as is generally seen only in neotheropods and ornithischians (Langer & Benton, 2006: character state 68-1; Nesbitt, 2011: character 269-1). In DMNH VP.63653, the process is not complete, but is also elongate and blade-like (Figs. 15A–15B). This differs from the slightly shorter preacetabular processes of Silesaurus (Dzik, 2003: fig. 11; Peecook et al., 2013: figs. 6F–6G), Ignotosaurus (Martinez et al., 2012) and especially from the extremely short and blunt process in Lutungutali (Peecook et al., 2013). In Eucoelophysis specimen GR 225 (Irmis et al., 2007a: fig. 2M), the process is incomplete. A preacetabular process that does not extend beyond the pubic peduncle is allegedly plesiomorphic for dinosauromorphs (Nesbitt, 2011: character state 269-0). Although Ferigolo & Langer (2007) claim this process is also short in Sacisaurus (MCN PV10100), it is mostly missing in that specimen (Langer & Ferigolo, 2013: fig. 10), and the ilium of Diodorus is undescribed. The highly elongate preacetabular process of Kwanasaurus is considered to be an autapomorphy.

The postacetabular process of DMNH EPV.48506, DMNH EPV.63653, and DMNH EPV.52195 (poa in Figs. 14–15) is large, slightly longer than the preacetabular process and extending well posterior to the acetabulum. It bears a large, ventrolaterally oriented brevis shelf sheltering a distinct brevis fossa (bs and bf in Figs. 14A–14B, 14G–14H, 15A and 15C) as occurs in other members of Sulcimentisauria: Silesaurus (Dzik, 2003), Eucoelophysis (GR 225; Irmis et al., 2007a: fig. 2M), Lutungutali (Peecook et al., 2013), Ignotosaurus (Martinez et al., 2012), and Sacisaurus (Langer & Ferigolo, 2013). A distinct brevis shelf and brevis fossa unites dinosaurs and some non-dinosaurian dinosauromorphs (Langer & Benton, 2006; Nesbitt, 2011), although both are weakly developed or absent in Asilisaurus (Nesbitt et al., 2010) and herrerasaurids (Langer & Benton, 2006). Very faint longitudinal striations occur along the lateral edge of the brevis shelf in all three DMNH specimens, but do not form the more rugose surface present in Silesaurus (Dzik, 2003: fig. 11), Ignotosaurus (Martinez et al., 2012) and Lutungutali (Peecook et al., 2013). The sharp ventrolateral edge of the brevis shelf merges with the a low rounded ridge that extends to the edge of the acetabulum (Figs. 14A–14B, 14G–14H, 15A and 15C), as in other silesaurids (Dzik, 2003; Ferigolo & Langer, 2007: fig. 2E; Irmis et al., 2007a: fig. 2M; Langer & Ferigolo, 2013: fig. 10a) and most other dinosauromorphs except theropods (Langer & Benton, 2006).

A small triangular process protrudes from the midpoint of the thin posteroventral edge of the postacetabular process of both DMNH EPV.48506 and DMNH EPV.63655 (Figs. 14A–14D and 15A–15B) which probably marked the posteroventral extent of the last (?third) sacral rib (Figs. 14C–14D; see below). A small similarly positioned projection is illustrated in Silesaurus (Dzik, 2003, Fig. 2), Ignotosaurus (Martinez et al., 2012: fig. 3), and Marasuchus lilloensis (Sereno & Arcucci, 1994b, fig. 6); this region is not well-preserved in DMNH EPV.52195, Sacisaurus (MCN PV10100; Langer & Ferigolo, 2013: fig. 10) or Eucoelophysis (GR 225; JW Martz, personal observation). In DMNH EPV.48506 and DMNH EPV.63653, a small foramen occurs near the edge of the brevis fossa, just anteroventral to the triangular process.

The acetabulum in both DMNH EPV.48506 and DMNH EPV.63653 (ac in Figs. 14A–14B, 14G–14H, 15A and 15C) is dorsoventrally deep with a well-developed and sharp-edged supracetabular crest (suc in Figs. 14–15), so that the acetabulum faces ventrally. As in Lutungutali (Peecook et al., 2013), there is no trace of an antitrochanteric fossa as occurs in Silesaurus (Dzik, 2003). In DMNH EPV.48506, the ventral edge of the ilium and acetabulum (the “ventral flange” of Martinez et al., 2012) is thin, ventrally concave, and seems to be a natural edge rather than a break (Figs. 14A–14D). In DMNH EPV.63653, the ventral rim of the acetabulum is clearly damaged, but the bone is extremely thin, suggesting that it had the same condition (Figs. 15A–15B). This suggests partial perforation of the acetabulum between the ilium and ischium/pubis. Nesbitt et al. (2010) considered a straight ventral margin of the acetabulum to be a silesaurid synapomorphy. If so, Kwanasaurus is the only known non-dinosaurian dinosauriform with a semiperforate acetabulum (Nesbitt et al., 2010; Nesbitt, 2011). Among archosaurs, an incompletely perforated acetabulum is an unusual feature but has been reported in Ornithosuchus longidens (Walker, 1964) and herrerasaurids (Langer & Benton, 2006).

The pubic and ischiac peduncles are both preserved in DMNH EPV.48506 and DMNH EPV.63653 (Figs. 14 and 15A–15C). The pubic peduncle (pup Figs. 14–15) is larger and bluntly truncated where it contacted the pubis. The pubic articulation is divided into an anteroventral facing surface and a more rugose ventrally facing surface (best seen in Figs. 14G–14H). The lateral margin of the pubic peduncle thins posterodorsally to become the supracetabular crest, and the medial margin tapers ventromedially to merge with the sharp ventral edge of the acetabulum. The ischial peduncle (isp in Figs. 14A–14B, 14G–14H and 15A–15C) is much smaller than the pubic peduncle and faces ventrolaterally.

On the medial side of the ilium in DMNH EPV.48506, the subhorizontal iliac blade forms a thin crest overhanging the rest of the medial surface (ilb in Figs. 14A and 15A–15C), extending between the anterior tip of the precetabular process to the posterior tip of the postacetabular process. The blade is not as well-preserved in DMNH EPV.52195 and DMNH EPV.63653 (Figs. 15B and 15E).

The regions of sacral rib attachment can be discerned in both DMNH EPV.48506 (Figs. 14C–14D) and DMNH EPV.63653 (Fig. 15B), although clear divisions between the attachments of different ribs are not clear, making an exact count impossible. The following interpretation of the sacral rib attachment sites, including the identification of primordial sacral ribs, is largely based on those made for other archosaurs (Novas, 1994: fig. 5B; Dzik, 2003: fig. 11B; Nesbitt, 2005: fig. 23C; Nesbitt, 2011: p. 117). The first primordial sacral rib probably attached in a slight depression on the anterior part of the medial surface of the ilium, just below the preacetabular process (sac 1.ar in Figs. 14D and 15B), while the second and possibly a third (primordial second?) sacral rib attached in a larger and more posterior depression (sac 2.ar and sac 3.ar in Figs. 14D and 15B) bounded dorsally by a short sharp-edge crest extending from the posterior margin of the postacetabular process, and posteroventrally by the small triangular projection on the thin posteroventral edge of the postacetabular process. These two depressions are connected over the acetabulum, and the entire region of sacral rib attachment is very faintly rugose. The rib attachment sites in Ignotosaurus appear to be very similar (Peecook et al., 2013: fig. 3F), although those authors only inferred the presence of two sacral ribs. Three or four ribs attach in Silesaurus (Dzik, 2003; Langer & Benton, 2006: p. 328; Nesbitt, 2011: p. 117) but the precise attachments are undescribed for other silesaurids for which the ilium is known.

Femur

Femora are by far the most common silesaurid elements from the Eagle Basin localities (Figs. 16–20; measurements in Table S2). The most complete is a large left femur (DMNH EPV.34579) from the Derby Junction locality (Fig. 16), but several proximal femora can also be assigned to Silesauridae: DMNH EPV.54828 (Figs. 17A–17E) and DMNH EPV.59311 (Figs. 19F–19J) from Shuvosaur Surprise, DMNH EPV.44616 (Figs. 17F–17J), DMNH EPV.56651 (Figs. 17K–17O), DMNH EPV.59301 (Figs. 19K–19O) from Main Elk Creek, DMNH EPV.63139 (Figs. 19A–19E) from Lost Bob, and DMNH EPV.63874 (Figs. 18F–18J) and DMNH EPV.125924 (Figs. 18A–18E) from Lost Bob East.

All of these specimen preserve at least two of the following silesaurid synapomorphies of the proximal end of the femur recognized in Asilisaurus, Silesaurus, Eucoelophysis, Sacisaurus, Diodorus, Lutungutali, and the Manda beds silesaurid (Dzik, 2003; Ferigolo & Langer, 2007; Nesbitt et al., 2010; Nesbitt, 2011; Kammerer, Nesbitt & Shubin, 2012; Langer & Ferigolo, 2013; Peecook et al., 2017; Barrett, Nesbitt & Peecook, 2015):The femoral head possesses a longitudinal groove in proximal view (gr in Figs. 17 and 19).

A flattened medial articular surface between the anteromedial and anterolateral tubers (amt and alt in Figs. 17, 18A–18B and 19).

A distinct notch ventral to the head (vn in Figs. 16–19).

As in all silesaurids except for Asilisaurus (Nesbitt et al., 2010; Nesbitt, 2011: character state 313-1), the proximal ends of these femora are also subtriangular in proximal view due to the absence of a well-developed posteromedial tuber (although a slight swelling is present at the same area in all Eagle Basin specimens) and a fossa trochanteris (=posterolateral depression, =facies articularis antitrochanterica).

Several femora that possess dinosauriform synapomorphies but are too badly damaged to preserve silesaurid synapomorphies have already been discussed. However, three distal femora, DMNH EPV.34028 from Main Elk Creek, DMNH EPV.59310 from Shuvosaur Surprise (neither figured), and DMNH EPV.67956 (Fig. 20; found in association with previously described scapula with the same number but too small to belong to the same individual) also do not preserve known silesaurid synapomorphies, but possess autapomorphies seen in the other Kwanasaurus femora (see below).

Nearly all specimens preserving the proximal end possess a distinct ridge-like dorsolateral trochanter (sensu Langer & Benton, 2006) on the proximal end of the femur (dt in Figs. 16–19), except for DMNH EPV.27699 and DMNH EPV.59311, where this region is damaged. The dorsolateral trochanter is best preserved in DMNH EPV.44616 (Figs. 17F–17G and 17J), DMNH EPV.59301 (Figs. 19K–19L and 19O), and DMNH EPV.63139 (Figs. 19A–19B). Although at least slightly damaged in the other specimens, the form seems to be consistent. The dorsolateral trochanter projects laterally from the shaft, sometimes curling slightly anterolaterally. Proximally, the trochanter thins and merges with the head. When well preserved, the posterolateral surface of the trochanter is somewhat flattened, bearing faint longitudinal grooves and ridges. The posterior margin of the proximal part of the femur is distinctly pinched into a rounded crest extending distally from the dorsolateral trochanter.

Nearly all specimens preserve a distinct anterior trochanter (=lesser or cranial trochanter) on the anterolateral surface of the femur, just distal to the head (at on Figs. 18–19). The anterior trochanter is an anteroposteriorly compressed crest extending parallel to the long axis of the femur. DMNH EPV.44616 is the only specimen with a perfectly preserved anterior trochanter, which is asymmetrically subtriangular in anterior and posterior views, slightly curled anterolaterally, and distinctly lacks a cleft between the trochanter and the main body of the femur (Figs. 17H and 17J); it is somewhat similar to the “longitudinal blade” forming part of the anterior trochanter of Silesaurus (Dzik, 2003: fig. 13). DMNH EPV.34579 (Fig. 16D), DMNH EPV.54828 (Fig. 17C), and DMNH EPV.125924 (Fig. 18C) possess a cleft between the trochanter and the main body of the femur, but it is not clear if this is natural or due to damage. The presence of an anterior (=lesser) trochanter is restricted to dinosauriforms and larger individuals of Dromomeron gregorii and Dromomeron gigas (Novas, 1992, 1996; Sereno & Arcucci, 1994b; Langer & Benton, 2006; Nesbitt et al., 2009a; Nesbitt, 2011; Martinez et al., 2015), while a cleft between the trochanter and the main body of the femur occurs in most theropods and some ornithischians (Novas, 1996; Langer & Benton, 2006) as well as the silesaurids Eucoelophysis, Sacisaurus, and Diodorus (Sullivan & Lucas, 1999: fig. 6; Ezcurra, 2006; Ferigolo & Langer, 2007; Kammerer, Nesbitt & Shubin, 2012). Peecook et al., (2017: p. 26) do not figure the anterior trochanter Lutungutali but describe it as as “mound-like,” suggesting that it is not cleft.

There is no trochanteric shelf (=transverse tuber sensu Dzik, 2003) in any of the Eagle Basin specimens except for DMNH EPV.125924, where a distinct scar interpreted as a weakly-developed trochanteric shelf (ts in Figs. 18B–18E) extends ventrolaterally from the anterior trochanter, resembling the trochanteric shelf in larger specimens of Dromomeron gregorii (Nesbitt et al., 2009a: figs. 2A–2B). In DMNH EPV.125924, the ventrolateral end of the trochanteric shelf ends with a posterolateral swelling that is present in other specimens lacking the shelf (sw in Figs. 16–18), and occurs in the same position as the end of the trochanteric shelf in Dromomeron romeri (Nesbitt et al., 2009a: fig. 2). The swelling is therefore interpreted as part of the attachment for the M. iliotrochantericus caudalis.

The trochanteric shelf is absent in known specimens of Sacisaurus, Eucoelophysis, Diodorus, and Lutungutali (Ferigolo & Langer, 2007; Ezcurra, 2006; Nesbitt et al., 2010; Kammerer, Nesbitt & Shubin, 2012; Langer & Ferigolo, 2013; Peecook et al., 2017), although it is present in Asilisaurus (Nesbitt et al., 2010), and some individuals of Silesaurus (Dzik, 2003; Piechowski, Tałanda & Dzik, 2014). The trochanteric shelf has been suggested to develop ontogenetically in at least some dinosauromorphs and to be highly subject to individual variation (Nesbitt, 2011; Griffin & Nesbitt, 2016a; Piechowski, Tałanda & Dzik, 2014). It should be noted however, that some specimens of Kwanasaurus lacking the shelf (most notably the largest and most complete specimen, DMNH EPV.34579) are similar in size to some of the larger femora of Silesaurus possessing the shelf (Dzik, 2003: fig. 13A; Piechowski, Tałanda & Dzik, 2014).

A fourth trochanter (ft in Figs. 16–19) is distinctly present in DMNH EPV.34579, DMNH EPV.63139, DMNH EPV.63874, DMNH EPV.125924, and the worn specimens DMNH EPV.43126, DMNH EPV.43588. The proximal end of the fourth trochanter rises smoothly from the posteromedial side of the femur as a pinched crest, distal to the distal end of the anterior trochanter on the opposite side of the femur. The trochanter is only fully preserved in DMNH EPV.125924, where it is proximodistally symmetrical (Figs. 18C–18E). The trochanter is also a low crest in specimens of Dromomeron romeri as well as Dromomeron gregorii (Nesbitt et al., 2009a), which are very different from the massive crest present in Ixalerpeton polesinensis (Cabreira et al., 2016: fig. S1D–S1E). At least in DMNH EPV.63874 (Figs. 18G–18H) and DMNH EPV.63139 (Figs. 19B–19C), where the region is well-preserved, a shallow depression occurs just anterior to the fourth trochanter on the medial side of the femur as in Diodorus (Kammerer, Nesbitt & Shubin, 2012) and Sacisaurus (Langer & Ferigolo, 2013).

The only femur with known silesaurid apomorphies for which the distal end of the femur is preserved is DMNH EPV.34579 (Fig. 16). The distal end is slightly expanded relative to the shaft. The sulcus dividing the medial and lateral condyles on the posterior side of the femur (Fig. 16B) extends about 1/3rd of the length of the shaft (sul in Fig. 16F), a silesaurid synapomorphy (Nesbitt et al., 2010). There is also a slight sulcus on the anterior side of the distal end (Figs. 16C–16D), causing the medial side of the distal end to protrude slightly anteriorly to the shaft. In distal view (Fig. 16B), the angle between the lateral condyle and the crista tibiofibularis (=fibular condyle) is obtuse, as in most archosaurs except for paracrocodylomorphs (Nesbitt, 2011).

The medial condyle (mc in Figs. 16B–16F) is a surprisingly sharp-edged flange, very similar to the crista tibiofibularis (tb in Figs. 16B–16F) in distal view, but smaller than both the crista tibiofibularis and lateral condyle (lc in Figs. 16B–16F). This appears to distinguish Kwanasaurus from Silesaurus, Diodorus, Sacisaurus, Eucoelophysis, and the large Manda beds silesaurid in which the medial condyle is quite broad and blunt in distal view (Sullivan & Lucas, 1999: fig. 5; Dzik, 2003; Kammerer, Nesbitt & Shubin, 2012: fig. 3E; Barrett, Nesbitt & Peecook, 2015: fig. 2). Indeed, this character state is shared uniquely between Kwanasaurus and lagerpetids (Nesbitt et al., 2010: character 225). There is a deep depression on the distal surface of the femur just anterior to the crista tibiofibularis (Fig. 16B); a depression also occurs on the distal end of the femur in Diodorus, but seems to occur between the medial condyle and crista tibiofibularis (Kammerer, Nesbitt & Shubin, 2012: fig. 3E).

Distal femora DMNH EPV.67956 (Fig. 20), DMNH EPV.34028, and DMNH EPV.59310 (the latter two specimens are not figured), do not possess known silesaurid apomorphies, and moreover the latter two specimens are somewhat worn. However, all three seem to share the autapomorphies present in DMNH EPV.34579: the medial condyle is at least slightly more slender and sharper-edged compared to both the lateral condyle and crista tibiofibularis, and a deep depression occurs on the distal surface behind the crista tibiofibularis (Fig. 20A). In DMNH EPV.67956, the sulcus between the medial condyle and crista tibiofibularis is a particularly deep groove (Figs. 20A and 20E).

Phylogenetic Analysis

Methods

Nesbitt’s (2011) phylogenetic analysis of Archosauriformes and Nesbitt et al.’s (2010) more focused analysis of Silesauridae have served as the basis for most subsequent analyses of silesaurids. The phylogenetic analyses of Kammerer, Nesbitt & Shubin (2012), Peecook et al. (2013) and Martinez et al. (2012), which described Diodorus, Lutungutali, and Ignotosaurus, respectively, all began with the data matrix of Nesbitt et al. (2010). The analyses of Langer & Ferigolo (2013), Bittencourt et al. (2014) and Agnolin & Rozadilla (2017) were all based on modified versions of the data matrix of Nesbitt (2011). Sarigül, Agnolin & Chatterjee (2018) modified the more recent data matrix of Nesbitt et al. (2017).

We have opted to utilize the data matrix of Peecook et al. (2013), acquiring the Nexus file from Morphobank. The matrix of Peecook et al. (2013) is slightly modified from the matrix of Nesbitt et al. (2010). Some characters have been renumbered to match the numberings given by Nesbitt et al. (2010) (see Appendix 2 for details). We further edited the Peecook et al. (2013) Nexus file in Mesquite (v. 3.51) by added the codings of Kwanasaurus williamparkeri from the present study, the codings of Diodorus scytobrachion from Kammerer, Nesbitt & Shubin (2012), the codings of Ignotosaurus fragilis from Martinez et al. (2012), the codings of Soumyasaurus aenigmaticus from Sarigül, Agnolin & Chatterjee (2018), and the codings of Technosaurus smalli based on our own observations. Additional codings were provided for Lutungutali sitwensis based on additional elements described by Peecook et al. (2017), for Eucoelophysis baldwini based on our own observations of the Ghost Ranch material, and for Dromomeron romeri based on the material described here. Due to Peecook et al. (2013) adding character 291 (shape of the shaft of the ischium) to the matrix of Nesbitt et al. (2010), the additional characters added by Kammerer, Nesbitt & Shubin (2012) to the same matrix were renumbered from 291 (tooth size through the dentary) and 292 (inclination of the dentary teeth) to 292 and 293, respectively. The character state for character 85 for Lewisuchus admixus was corrected by changing it from “1” to “?” (the dentary is unknown for that taxon). We also added a new character, the presence or absence of a longitudinal ridge on the dentary (character 294). The codings of Lewisuchus admixus/Pseudolagosuchus major were combined following previous workers (Nesbitt et al., 2010; Peecook et al., 2013). These modifications brought the total number of taxa in the analysis to 39, and the total number of characters to 294.

As nearly all silesaurid elements from the Eagle Basin are individual elements, the codings for Kwanasaurus are a composite of multiple specimens (Appendix 2). Moreover, the dinosauriform scapula and tibiae described above, which are potentially silesaurid but lack known synapomorphies for the clade, are also included in the composite. Although such compositing is not ideal, it has been used by other researchers (Kammerer, Nesbitt & Shubin, 2012; Langer & Ferigolo, 2013) and is difficult to avoid given that silesaurids are often recovered as individual elements (Irmis et al., 2007a; Kammerer, Nesbitt & Shubin, 2012: p. 278; Langer & Ferigolo, 2013: p. 355; Martinez et al., 2012), with only some taxa being known from associated elements (Dzik, 2003; Nesbitt et al., 2010; Peecook et al., 2013; Bittencourt et al., 2014).

We conducted our analysis using PAUP 4.0a165 for Macintosh OS. Following Nesbitt et al. (2010) and Peecook et al. (2013), all characters were equally weighted and characters 23, 78, 89, 98, 116, 142, 159, 169, 175, 177, 195, 200, 227, 250, and 281 were ordered. Erythrosuchus africanus and Euparkeria capensis were chosen as paraphyletic outgroups. Trees were searched for using the parsimony criterion implemented under the heuristic search option on Wagner trees using tree bisection–reconnection (TBR) branch-swapping with 1,000 random addition sequences holding 10 trees per replicate, continuing subsequent TBR swapping on all stored minimum length trees. Using these criteria, separate analyses were conducted both including and excluding Ignotosaurus, Technosaurus, and Soumyasaurus. These three taxa had the lowest number of identifiable character states, and are known from the fewest number of elements (the ilium in Ignotosaurus, and only the dentary in Technosaurus and Soumyasaurus).

Results

In the following discussion, clade definitions were taken from Langer et al. (2013) and sources cited therein, except for the new clade name Sulcimentisauria introduced here (see Systematic Paleontology). Our analysis including Ignotosaurus, Technosaurus, Soumyasaurus recovered 72 most parsimonious trees (MPTs) with a best score tree length of 776 (C.I = 0.464, R.I. = 0.707). The strict consensus tree (Fig. 23A) collapses nearly all silesaurid taxa into a polytomy with sauropodomorphs and ornithischians. The Adams consensus tree (Fig. 23B) does little better; Ignotosaurus fragilis forms a polytomy with Silesauridae and Dinosauria, making the status of Ignotosaurus as a silesaurid unclear. However, as with most previous analyses (Nesbitt et al., 2010; Nesbitt, 2011; Kammerer, Nesbitt & Shubin, 2012; Martinez et al., 2012; Peecook et al., 2013; Sarigül, Agnolin & Chatterjee, 2018), the Adams consensus tree found Lewisuchus admixus/Pseudolagosuchus major, Soumyasaurus aenigmaticus, and Asilisaurus kongwe to be consecutive outgroups to Sulcimentisauria, which contains all other silesaurids. Within Sulcimentisauria, Silesaurus opolensis was found to be sister taxon to all other sulcimentosaurians. Interestingly, the two African taxa (Lutungutali sitwensis and Diodorus scytobrachion) are sister taxa, as are the two taxa from the Chinle Formation of western North America (Eucoelophysis baldwini and Kwanasaurus williamparkeri). However, these two clades form a polytomy with Sacisaurus agudoensis and Technosaurus smalli. Pisanosaurus mertii, which was found to be another basal silesaurid by Agnolin & Rozadilla (2017), was recovered as an ornithischian.

Figure 23 Phylogenetic analysis of Silesauridae.

(A) Strict consensus tree for phylogenetic analysis incorporating Ignotosaurus, Technosaurus, and Soumyasaurus. (B) Adams consensus tree for same phylogenetic analysis. (C) Identical strict consensus and Adams consensus trees for phylogenetic analysis excluding Ignotosaurus, Technosaurus, and Soumyasaurus.

Re-running the analysis without the problematic taxa Ignotosaurus, Technosaurus, and Soumyasaurus recovered six MPTs with a best score tree lengths of 772 (C.I = 0.465, R.I. = 0.706). The removal of Ignotosaurus allowed the strict consensus tree and identical Adams consensus tree (Fig. 23C) to resolve Silesauridae as sister taxon to Dinosauria as in most previous analyses (Nesbitt et al., 2010; Nesbitt, 2011; Kammerer, Nesbitt & Shubin, 2012; Martinez et al., 2012; Peecook et al., 2013; Sarigül, Agnolin & Chatterjee, 2018). Otherwise the topology was similar to that of the analysis including the problematic taxa, with Lewisuchus admixus/Pseudolagosuchus major and Asilisaurus kongwe being consecutive sister taxa to Sulcimentisauria, Silesaurus opolensis being sister taxon to all other sulcimentisaurians, and the African taxa (Lutungutali sitwensis and Diodorus scytobrachion) and North American taxa (Eucoelophysis baldwini and Kwanasaurus williamparkeri) forming clades in a polytomy with Sacisaurus aguodensis. Synapomorphies for clades in the strict consensus/Adams consensus trees excluding the problematic taxa are given in Appendix 3.

Discussion

Within the last decade, it has become clear that the Late Triassic dinosaur assemblage of western North America was of low diversity, being represented only by basal theropods and basal neotheropods that co-existed with non-dinosaurian dinosauromorphs (lagerpetids and silesaurids) (Nesbitt, Irmis & Parker, 2007, Nesbitt et al., 2009a, 2009b; Irmis et al., 2007a; Sues et al., 2011; Marsh et al., 2016; Marsh, 2018). While the western North American Late Triassic dinosauromorph fauna has been previously described from the Colorado Plateau and western Texas (Ezcurra, 2006; Nesbitt, Irmis & Parker, 2007; Irmis et al., 2007a; Nesbitt & Chatterjee, 2008; Martz et al., 2013), the Eagle Basin dinosauromorph fauna described here for the first time in detail demonstrates that similar patterns of dinosauromorph diversity existed north of the Ancestral Uncompahgre Highlands. Indeed, the Eagle Basin fauna is the northernmost Triassic dinosauromorph fauna known from North America with the possible exception of basal neotheropod material from the Nugget Sandstone in Utah, which might be Upper Triassic or Lower Jurassic (Britt et al., 2010, 2015). However, unlike the Utah material, the Eagle Basin collection includes lagerpetids and silesaurids, giving it the northern-most occurrence of non-dinosaurian dinosauromorphs in North America. Coelophysoid neotheropods are also known from the Eagle Basin, and will be described in a future publication. Although non-neotheropod theropods such as Tawa (Nesbitt et al., 2009b), Daemonosaurus (Sues et al., 2011) and Chindesaurus (Long & Murry, 1995; Marsh et al., 2016) have not been identified in the northern Colorado assemblage, much material from the Eagle Basin localities remains to be prepared.

Size and morphological variation within Kwanasaurus

A tentative composite skeleton reconstruction for Kwanasaurus williamparkeri is presented in Fig. 24. Composited from multiple elements of different sizes, the reconstruction is based on the highly ambiguous assumption that Kwanasaurus was proportioned like Silesaurus, with the scale bars representing the smallest and largest femora in the quarry. This size variation is best illustrated by the femora, the most commonly encountered element (Table S2). The largest preserved femur (DMNH EPV.34579; Fig. 18) is about 18 cm long, while the smallest (DMNH EPV.63139; Figs. 19A–19E) is estimated by comparison to have been perhaps six cm long.

Figure 24 Reconstruction of Kwanasaurus williamparkeri.

(A) Skeletal reconstruction with elements based on individuals of varied sizes, all scaled under the assumption that Kwanasaurus is proportioned similarly to Silesaurus. (B) Life reconstruction. Scale bars = 10 cm, given for probable largest specimen (DMNH EPV.34579) and one of the smallest specimens (DMNH EPV.63139).

As with Silesaurus (Dzik, 2003; Piechowski, Tałanda & Dzik, 2014), Asilisaurus (Griffin & Nesbitt, 2016a), and Lutungutali (Peecook et al., 2017), Kwanasaurus has a large sample size of femora spanning a range of sizes, giving possible insights into ontogenetic changes. Assuming that this size variation is largely ontogenetic, qualitative examination of the material shows few obvious morphological changes with ontogeny, although most elements are at least partially damaged and few are even close to complete. Therefore, little can be said with confidence. However, it is worth noting that development of the muscle attachments does not seem to be subject to strong variation as occurs in Asilisaurus and theropods (Griffin & Nesbitt, 2016a, 2016b). In particular, the lesser trochanter in Kwanasaurus is a simple, longitudinally oriented process with no trochanteric shelf except for DMNH EPV.125924, where the trochanteric shelf is present but weakly developed.

Interesting differences do occur between the large holotype maxilla (DMNH EPV.65879; Figs. 8A–8H) and the smaller referred specimens (DMNH EPV.63650, DMNH EPV.125921, and DMNH EPV.125923; Figs. 8I–8P and 9). All maxillae are relatively robust elements compared to other silesaurids, and possess fused dentition and the enormous medial flange characterizing the taxon. However, the smaller specimens do not possess the prominent sutural surfaces for the jugal, lacrimal, and palatine seen in the larger holotype, so these may have developed with increased maturity.

The distinctiveness of Kwanasaurus from other North American silesaurids

Kwanasaurus williamparkeri contributes to our understanding of North America silesaurid diversity. It is the fourth silesaurid alpha taxon named from North America following Eucoelophysis baldwini (Sullivan & Lucas, 1999; Ezcurra, 2006; Nesbitt, Irmis & Parker, 2007; Breeden et al., 2017), Technosaurus smalli (Chatterjee, 1984; Nesbitt, Irmis & Parker, 2007; Martz et al., 2013) and Soumyasaurus aenigmaticus (Sarigül, Agnolin & Chatterjee, 2018). Assuming that all elements discussed here truly belong to the same taxon, Kwanasaurus is currently the most thoroughly described North American silesaurid.

Kwanasaurus seems to be distinct from Eucoelophysis baldwini. The two taxa share leaf-shaped denticulate teeth and a ventrally placed Meckelian groove, but these occur in other sulcimentisaurians. Perhaps more significantly, both taxa have a pronounced lateral ridge on the dentary, a feature shared with Diodorus (Figs. 22A, 22E and 22M). However, Kwanasaurus possesses character states absent in Eucoelophysis: a highly elongate and bladelike preacetabular process of the ilium (Figs. 14–15), a relatively small and slender medial distal condyle of the femur compared to lateral condyle and crista tibiofibularis, and a depression on distal end of the femur anterior to the crista tibiofibularis (Figs. 16B and 20B). Moreover, Eucoelophysis autapomorphically lacks a fourth trochanter (Breeden et al., 2017), which is present in Kwanasaurus (Figs. 16–20).

The taxonomic distinctiveness of Kwanasaurus from the holotype and only known specimen of Technosaurus smalli is more ambiguous as the latter specimen is currently accepted to include only the dentary and premaxilla (Nesbitt, Irmis & Parker, 2007; Martz et al., 2013), which are both poorly preserved; other elements assigned to the taxon by Chatterjee (1984) have been re-identified as shuvosaurid and theropod (Irmis et al., 2007b; Nesbitt, Irmis & Parker, 2007). As the premaxilla is not known in Kwanasaurus, this permits only the dentaries to be compared. Technosaurus seems to lack the lateral ridge on the dentary shared by Kwanasaurus, Eucoelophysis, and Diodorus, and the dentary teeth of Technosaurus, though damaged, appear to be somewhat more robust than those of Kwanasaurus (Figs. 22G–22H). We therefore tentatively consider Kwanasaurus and Technosaurus to also be distinct taxa.

Soumyasaurus aenigmaticus is known from a single incomplete dentary (Figs. 22O–22P; Sarigül, Agnolin & Chatterjee, 2018). The dentary of Soumyasaurus is extremely slender compared to that of Kwanasaurus, the anterior part is anteroventrally oriented as in Asilisaurus whereas that of Kwanasaurus is anterodorsally oriented, and it seems to lack a lateral ridge present in Kwanasaurus (Sarigül, Agnolin & Chatterjee, 2018: fig. 5). Moreover, the tooth crowns of Soumyasaurus are small and conical, whereas the crowns of Kwanasaurus are broad and denticulate.

North American silesaurid biochronology

The age of Kwanasaurus relative to the other three western North American taxa is unclear. Technosaurus and Soumyasaurus are known from the Post Quarry vertebrate assemblage in the lower Cooper Canyon Formation of the Dockum Group in Texas (Chatterjee, 1984; Martz et al., 2013; Sarigül, Agnolin & Chatterjee, 2018), which on the basis of lithostratigraphic correlation and the overall nature of the assemblage, probably falls within the later part of the Adamanian estimated holochronozone (Martz et al., 2013), with a plausible late Lacian or early Alaunian age between 220 and 215 Ma (Martz & Parker, 2017). The Hayden Quarry, which lies in the Mesa Montosa Member or lower Petrified Forest of the Chinle Formation (Lucas et al., 2003; Irmis et al., 2007a), contains silesaurid material assigned to Eucoelophysis (Irmis et al., 2007a; Breeden et al., 2017) that falls within the early part of the Revueltian estimated holochronozone (Martz & Parker, 2017), making it slightly younger than the Post Quarry. The Hayden Quarry is very well-constrained geochronologically by a radiometric date of 211.9 ± 0.7 Ma (Irmis et al., 2011), making it late Alaunian in age. The postulated Revueltian age for Kwanasaurus suggests that it is at least closer in age to Eucoelophysis than to Technosaurus and Soumyasaurus.

Silesaurid phylogeny and distribution

Silesaurids were herbivorous non-dinosaurian dinosauriforms that lived during the Middle and Late Triassic (Ladinian-Norian) and had a cosmopolitan distribution across both the northern and southern regions of Pangea (Nesbitt et al., 2010; Langer et al., 2013). They are represented by at least 11 putatively acknowledged alpha taxa, including the four from North American already discussed (Fig. 25).

Figure 25 Global and temporal distribution of non-dinosaurian dinosauromorphs.

(A) Lagerpetid distribution, (B) Silesaurid distribution.

The consecutive sister taxa to Sulcimentisauria, as well as the broad taxonomic composition of Sulcimentisauria itself, has been fairly consistent in phylogenetic analyses (Nesbitt et al., 2010; Nesbitt, 2011; Kammerer, Nesbitt & Shubin, 2012; Martinez et al., 2012; Peecook et al., 2013; Sarigül, Agnolin & Chatterjee, 2018), including the present study. As the inclusion of Soumyasaurus and Technosaurus did nothing to change the relationships of other taxa within the analyses, they are likely sulcimentisaurians as found in the strict consensus and Adams consensus analyses that included them. However, unlike Martínez et al. (2013), we were not able to confirm the placement of Ignotosaurus within Sulcimentisauria, or even Silesauridae (Figs. 23A–23B). Within Sulcimentisauria, Silesaurus opolensis was consistently found to be sister taxon to all other sulcimentisaurians. Interestingly, there are indications of African and North American clades within Sulcimentisauria, although the topology of this part of the tree is not well resolved.

This broad picture of silesaurid evolution is somewhat geochronologically consistent. The possibly synonymous Lewisuchus and Pseudolagosuchus are not only the basal-most silesaurids, but among the oldest known, occurring in the early Carnian (Late Triassic) Chañares Formation of Argentina (Bittencourt et al., 2014; Marsicano et al., 2016). Asilisaurus kongwe, the sister taxon to Sulcimentisauria in both strict consensus and Adams consensus trees, is slightly older, being known from the Anisian (Middle Triassic) of Tanzania (Nesbitt et al., 2010). The youngest possible non-sulcimentisaurian silesaurid is Soumyasaurus aenigmaticus (Sarigül, Agnolin & Chatterjee, 2018) from the early Norian (Late Triassic) of Texas.

In contrast, all sulcimentisaurians (Lutungutali sitwensis, Eucoelophysis baldwini, Technosaurus smalli, Kwanasaurus williamparkeri, Sacisaurus aguodensis, Silesaurus opolensis, and Diodorus scytobrachion) are Late Triassic in age (Dzik, 2003; Irmis et al., 2007a; Martz et al., 2013; Langer & Ferigolo, 2013; Martinez et al., 2012; Peecook et al., 2017). Given the extremely small size of Soumyasaurus relative to most other silesaurids (Sarigül, Agnolin & Chatterjee, 2018), it is tempting to speculate that it may be a juvenile of a Late Triassic sulcimentisaurian (possibly Technosaurus from the same locality) with its apparently plesiomorphic character states being ontogenetic. If this is the case, then all known non-sulcimentisaurian silesaurids are Ladinian to early Carnian in age, and sulcimentisaurians range from Carnian to Norian in age.

In summary, phylogenetic analysis and current age estimates suggests an Early or Middle Triassic origin for Silesauridae in southern Gondwana. Sulcimentisauria expanded from Gondwana into Laurasia during the Late Triassic, becoming established in Europe (Silesaurus), South America (Sacisaurus), Africa (Lutungutali and Diodorus), and North America (Eucoelophysis, Kwanasaurus, and probably Technosaurus) from the Carnian to at least as late as the Norian.

Silesaurid paleoecology

Herbivorous dinosaurs originated during the Carnian stage and became dominant herbivores during the Norian in the higher latitudes (Langer et al., 2010). However, in the lower-mid latitude Norian Chinle/Dockum beds of the western United States, herbivorous dinosaurs appear to have been absent (Nesbitt, Irmis & Parker, 2007; Irmis et al., 2011). Instead, other amniotes have been identified as possibly occupying herbivorous or omnivorous niches, including Trilophosaurus (Gregory, 1947), shuvosaurids (Nesbitt, 2007), Revueltosaurus (Heckert, 2002; Parker et al., 2005), aetosaurs (Desojo et al., 2013), archosauriforms of uncertain affinity with ornithischian-like teeth (Heckert, 2002; Nesbitt, Irmis & Parker, 2007), and dicynodonts (Camp & Welles, 1956).

Silesaurids were also widely distributed Late Triassic herbivores in both wet and dry climate belts of various latitudes all over the world (Langer et al., 2013), including tropical western North America. While the remains of silesaurids are known from multiple locations and stratigraphic levels within the Chinle Formation and Dockum Group, they are generally rare (Martz et al., 2013; Parker, Irmis & Nesbitt, 2006; Ezcurra, 2006; Nesbitt & Chatterjee, 2008), except for the Hayden Quarry in New Mexico (Irmis et al., 2007a; Breeden et al., 2017) and the Eagle Basin (this study) where their remains are locally abundant.

An overview of silesaurid dental diversity suggests that their widespread distribution across Pangea may have been driven, at least in part by their dietary adaptability. Lewisuchus admixtus, the sister taxon to all other silesaurids, retained the probably plesiomorphic slender jaws and ziphodont dentition of other early dinosauromorphs and theropods (Bittencourt et al., 2014), suggesting a faunivorous diet. In contrast, the more derived Asilisaurus kongwe (Nesbitt et al., 2010), Soumyasaurus aenigmaticus (Sarigül, Agnolin & Chatterjee, 2018), and Silesaurus opolensis (Dzik, 2003; Kubo & Kubo, 2014) had almost conical teeth with weakly developed serrations coincident with the development of an edentulous beak on the lower jaw (Appendix 3); patterns of microwear on the teeth of Silesaurus suggest that it was herbivorous or omnivorous (Kubo & Kubo, 2014). In contrast, most members of Sulcimentisauria possessed short and broad folidont teeth (sensu Hendrickx, Mateus & Araújo, 2015) with massive denticles, similar to those of predominantly herbivorous reptiles (Reisz & Sues, 2000; Barrett, 2000). The basal placement of Silesaurus within Sulcimentisauria is consistant with this pattern, suggesting that the earliest members of the clade still possessed conical teeth, but that folidont dentition evolved prior to the radiation of sulcimentisaurians into South America, Africa, and North America.

The overall picture of silesaurid dental evolution suggests a shift from faunivorous to increasingly herbivorous species throughout the Triassic. Ziphodont-toothed taxa and taxa with conical teeth were restricted to Gondwana during the Anisian and Carnian, and eventually by sulcimentisaurian taxa with predominantly folidont teeth that radiated across both Gondwana and Laurasia in the Carnian and Norian. These stages mirror the development of herbivorous dietary specialization in sauropodomorphs that also occurred during the Late Triassic (Barrett, Butler & Nesbitt, 2011: p. 386) reinforcing the evidence for convergent evolution among herbivorous dinosauromorphs (Barrett, Nesbitt & Peecook, 2015).

Kwanasaurus is suggested here to bear the most extreme adaptations for folivory yet known within Silesauridae. In addition to possessing leaf-shaped denticulate teeth, their maxilla is an extremely short and robust element compared to the more slender maxillae of other silesaurids (Fig. 21), with thick, almost durophagous folidont teeth, and extremely prominent sutural surfaces for contact with the palatine, jugal, and lacrimal on a massive flange unlike anything seen in other silesaurid taxa. The dentary does not seem to have been as massive, but is at least more robust than the extremely slender elements in Eucoelophysis, Sacisaurus, and Soumyasaurus (Fig. 22). The development of a longitudinal ridge on the dentary in Kwanasaurus and some other sulcimentisaurians (Appendix 3) may also be related to reinforcing the lower jaw. These adaptations suggest that Kwanasaurus had a relatively powerful bite in which the maxilla was reinforced by strong contacts with other skull elements. The taxon may therefore have been consuming tougher food than most other silesaurids, similar to the tendency of herbivorous lizards to evolve more compact and powerful skulls to deal with tough, fibrous plant material (Metzger & Herrel, 2005).

Supplemental Information

Supplemental Information 1 Supplemental Appendices.

Click here for additional data file.

Supplemental Information 2 Figure S1. Measurements of appendicular elements detailed in Appendix 1.

(A) Dromomeron romeri proximal femur in proximal view, (B) posteromedial view, (C) posterolateral view. (D) Dromomeron romeri humerus in proximal view, (E) anterior view, (F) medial view, (G) distal view. (H) Dinosauriformes scapula in lateral view, (I) posterior view, (J) ventral view. (K) Dinosauriformes tibia in proximal view, (L) lateral view, (M) posterior view, (N) distal view. (O) Silesauridae humerus in proximal view, (P) anterior view, (Q) medial view, (R) distal view. (S) Silesauridae femur in proximal view, (T) anteromedial view, (U) anteromedial view, (V) distal view.

Click here for additional data file.

Supplemental Information 3 Supplemental_Table_S1_Teeth.

Click here for additional data file.

Supplemental Information 4 Supplemental_Table_S2_Appendicular_measurements.

Click here for additional data file.

Supplemental Information 5 Supplemental File: Modified Peecook et al. (2013) Nexus file.

Click here for additional data file.

We thank editor Hans Dieter-Sues and reviewers Brandon R. Peecook and Max Langer for extremely helpful reviews that improved the quality of the manuscript. We also thank Bill Parker and Sterling Nesbitt for useful discussions. Randy Irmis, Bill Mueller, and Sterling Nesbitt provided us with photos of Technosaurus, Eucoelophysis, Silesaurus, and Asilisaurus. Special thanks to numerous DMNH volunteers who have assisted in the field excavations and preparation of much of the material presented here. Denver Museum of Nature and Science provided access to specimens in their collection. Ben Creisler’s linguistic research and advice was invaluable in formulating the taxon names Kwanasaurus and Sulcimentisauria. Susan Drymala was invaluable in guiding us through the use of PAUP and Mesquite.

Anatomical Abbreviations

ac acetabulum

afe antorbital fenestra

afo antorbital fossa

ag articular glenoid

alt anterolateral tuber

amp anteromedial process

amt anteromedial tuber

an angular

an.ar articulation with the angular

as.a articular surface for the ascending process of the astragalus

asc apex of scapula

asm ascending process of the maxilla

at anterior trochanter

bf brevis fossa

bs brevis shelf

brk broken bone surface

cc cnemial crest

cnc concavity

cnv convexity

co.ar articulation with the coracoid

dc deltopectoral crest

dt dorsolateral trochanter

ec ectotuberosity

ect ectepicondyle

en entotuberosity

ent entepicondye

ecf ectepicondylar flange

faa facies articularis antitrochantera

fc fibular crest

fo foramen

ft fourth trochanter

gr groove

ilb iliac blade

isp ischial peduncle

ju.la.ar jugal and lacrimal articulation

lc lateral condyle

lr lateral ridge

mc medial condyle

maf mandibular fenestra

mef medial flange

Mk Meckelian groove

mt medial tuberosity

pa.ar palatine articulation

pit pit

pra preacetabular process

poa postacetabular process

plf posterolateral flange

plp posterolateral process

pm.ar premaxilla articulation

pmf promaxillary fossa

pmt posteromedial tuber

pup pubic peduncle

rf replacement foramina

rp replacement pits

rt root

sa.ar articular surface for the surangular

sac #.ar articulation for sacral #

suc supracetabular crest

sul sulcus

sw swelling

sy symphysis

tb crista tibiofibularis

tc thin crest

ts trochanteric shelf

ve ventral emargination

vn ventral notch

vo.ar vomerine flange.

Institutional Abbreviations

DMNH EPV Denver Museum of Nature and Science, Denver, Colorado, USA

GR Ghost Ranch Ruth Hall Museum of Paleontology, Ghost Ranch, New Mexico, USA

MCN PV Museu de Ciências Naturais, Fundação Zoobotanânica do Rio Grande do Sul, Porto Alegre, Brazil.

MHNM-ARG Museum d’Histoire Naturelle de Marrakech (Argana Basin Collection), Marrakech, Morocco

TMM Texas Memorial Museum, Austin, Texas, USA

TTU P Museum of Texas Tech University Paleontology, Lubbock, Texas, USA

ZPAL Institute of Paleobiology of the Polish Academy of Sciences in Warsaw, Poland.

Additional Information and Declarations

Competing Interests

Author Contributions

Field Study Permissions

Data Availability

New Species Registration

The authors declare that there are no competing interests.

Jeffrey W. Martz conceived and designed the experiments, performed the experiments, analyzed the data, contributed reagents/materials/analysis tools, prepared figures and/or tables, authored or reviewed drafts of the paper, approved the final draft.

Bryan J. Small conceived and designed the experiments, performed the experiments, analyzed the data, contributed reagents/materials/analysis tools, authored or reviewed drafts of the paper, approved the final draft, collected and supervised preparation of material.

Field permits provided and approved by the United States Bureau of Land Management (permit numbers C-49819 and C-49819d).

The following information was supplied regarding data availability:

Morphobank Project DOI: 10.7934/P3508.

The following information was supplied regarding the registration of a newly described species:

Publication LSID: urn:lsid:zoobank.org:pub:20FCEEA6-4512-42FD-BAE9-A570BF4611F4

Kwanasaurus LSID: urn:lsid:zoobank.org:act:E9514954-F9FD-4D79-A620-D705122D59D5

Kwanasaurus williamparkeri LSID: urn:lsid:zoobank.org:act:25A4AE71-56B3-4797-B30D-1FA1D37E1F3F.

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
