# Peer review of "Non-dinosaurian dinosauromorphs from the Chinle Formation (Upper Triassic) of the Eagle Basin, northern Colorado: Dromomeron romeri (Lagerpetidae) and a new taxon, Kwanasaurus williamparkeri (Silesauridae)"

_PeerJ, doi:10.7717/peerj.7551_

## Round 0.1 · original submission · Minor Revisions

Both referees have lists of minor changes and corrections, all of which should be easily addressed.

·

Basic reporting

The authors do an excellent job clearly reporting the presence of multiple dinosauromorphs from the Eagle Basin of Colorado. A new genus and species of silesaurid is erected and a commonly recovered clade with Silesauridae is given an appropriate taxon name. The taxonomy is well done and the chosen names are quite good.

LIT CITED:
This work integrates much of our current understanding of Triassic Dinosauromorpha, but I would suggest a few additions.
1) Peecook et al. 2018 (SVP Memoir): Peecook et al. follow the same logic as the paper under review in referring morphologically consistent silesaurid elements from the Ntawere Formation to the one named taxon (Lutungutali sitwensis). Like in the Eagle Basin, silesaurid elements in Zambia are found isolated. I’d encourage the authors to consider adding comparisons of the maxilla (Line 891) and femur of Lutungutali to the descriptions of Kwanasaurus. I applaud the authors for the clarity with which they explain the rationale of referring all silesaurid bones to Kwanasaurus, and the openness with which they address the potential shortcomings of the assumption. Line 470.

Peecook et al. also describe a range of size variation in Ntawere silesaurid remains (femora from ~60 to ~380mm). Lines 1303-1310.

Probably the most important contribution of Peecook et al. is the explicit treatment of the lack of age constraints on ‘Middle’ Triassic assemblages in Zambia and Tanzania. At this point, radiometric dates from South America override now defunct long range biostratigraphy to indicate that both the Zambian and Tanzanian assemblages are at least Ladinian and quite possibly Carnian (Late Triassic) in age. No evidence supports the statement that the Zambian or Tanzanian assemblages are Anisian in age. This changes the overview written by the authors, making the story simpler. Lines 1379-1393.

2) Barrett et al. 2015 (Gondwana Research): Barrett et al. describe a very large silesaurid femur (~345mm) from the Lifua Member of the Manda Beds in Tanzania. At the time of publication, the large femur was hypothesized to be a somatically mature individual of Asilisaurus. Lines 1303-1310.

3) Marsh 2018 (PaleoBios): This paper is cited by the authors, but I would also include it on line 1277 at the opening of the Discussion.

4) Wynd et al. 2018 (SVP Memoir): Wynd et al. reinforces the age constraints for Zambian and Tanzanian Triassic assemblages using the therapsid record (Cynognathus, Diademodon, Kannemeyeria).

FIGURES:
The stereo photos and accompanying line drawings are excellent.

Figure 11 E&F: the angular is mislabeled as the articular ‘ar’.
Figure 11F: ‘rf’ is incorrectly written as ‘re’ and there are no guidelines. If ‘re’ is not a mistake, it is not in the Anatomical Abbreviations.
Figure 13 B&J&N: ‘rf’ is incorrectly written as ‘re’.
Figures 19, 21: ‘lg’ is not in the Anatomical Abbreviations.
Figure 20: should ‘ts’ be ‘tc’?

Experimental design

The figures and descriptions are thorough, and the appendices contain additional comprehensive morphological data. Any paleontologist could recreate the analyses of the authors easily.

Validity of the findings

Strong discussion. I expect interesting future work will be possible on the dentition and trophic evolution of Silesauridae.

Additional comments

Line 183: It is unclear is the associated pseudosuchian elements are part of the specimen DMNH EPV.63873 as currently written.
Line 194: 'ends of the femora form'
Line 1107: Should the femoral head feature be the first of the listed autapomorphies, and therefore '1'?

·

Basic reporting

The MS by Martz & Small reports important new findings from a relatively poorly explored outcropping area of the Chinle Formation – the Eagle Basin, in Colorado – describing new dinosauromorph specimens, including a new silesaurid taxon – Kwanasaurus williamparkeri. Hence, it provides data for a better understanding of Triassic diversity at higher latitudes of the northern hemisphere. The MS is well executed, does not have significant flaws, and can be published nearly as it is. I would just ask the authors to consider the minor points listed in the "General comments for the author".

Experimental design

See "Basic reporting" and "General comments for the author"

Validity of the findings

See "Basic reporting" and "General comments for the author"

Additional comments

Lines 38, 459, and 1266: it is preferable to avoid using the term “more derived” in phylogenetic studies, as one may wonder in which direction is that “derivation”. So, replace “more derived than” by (for example) “forming the sister group to”
Line 39: replace “specialization” by “specializations”
Lines 48, 64, 72, and 1381: it is important to note that the Chañares Formation has been radioisotopically dated as early Carnian (Late Triassic) so that its fossil taxa can no longer be considered representatives of the Middle Triassic fauna (Marsicano et al. 2016. Proc. Nat. Acad. Sci. USA. 509–513). In line 48, the Middle Triassic reference may be correct as a historical account, but this is not the case in lines 64,72, and 1381.
Line 50: with a single putative record (Pisanosaurua mertii) ornithischians can hardly be considered to have a “global distribution”. Please rewrite accordingly.
Line 60: which work? There are two paper mentioned beforehand.
Line 75: also during Norian times, as indicated by Sacisaurus agudoensis, in the Caturrita Formation, and Dromomeron gigas, in the Los Colorados Formation.
Line 159 and along the text: strictly speaking, autapomorphies apply only to species. Hence, replace “autapomorphy(ies)” by “apomorphy(ies)” in lines 159, 161, 193, 257, 260, 273, 324, 810, 892, 973, 1104, 1122, and 1238. As for lines 388, 389, and 1023, even if Kwanasaurus is a monospecific genus and the generic epithet is used as the reference for the species, the term autapomorphy should still be avoided, because Kwanasaurus is a supraspecific taxon (also for other genera in lines 454 and 1334)
Line 195: replace “ventrally” by “distally”
Line 209: replace “autapomorphy” by “autapomorphies”
Line 222: include “in” before “shuvosaurids”
Line 234: replace “shaft” by “bone”
Line 256: include “,” after “Marasuchus”
Line 261: replace “The” by “That”
Line 276: delete “the” before “much”
Line 278: delete “In ventral view,”
Line 280: replace “element” by “scapula”
Line 283: replace “surface” by “margin” (two times)
Line 289: include “groups” after “two”
Line 292: replace “distal” by “dorsal”
Line 294: delete “occurs” before “in Tawa”
Lines 298-299: replace “This may indicate an ossified suprascapula” by “This may indicate that an ossified suprascapular was present”
Line 315: replace “and likely belong to Kwanasaurus, but” by “, which likely belong to Kwanasaurus. Yet,”
Line 325: replace “posterior lateral and medial” by “lateral and medial posterior”
Line 329: “more or less” should be replaced by a more precise term.
Lines 333-334: replace “lacks a concavity separating the crest from the condyles” by “is not separated from the condyles by a concavity”
Line 344: replace “and then become” by “then becoming”
Line 346: replace “at” by “of”
Line 352: explain better the “distal thickening”
Line 359: include “a” after “crest,”
Line 361: replace “The elements cannot currently be completely ruled out as” by “Yet, it cannot be completely ruled out that the elements correspond to”
Lines 368-370: definition of Silesauridae is unnecessary, exclude.
Line 388: include “all” before “other”
Lines 398-421: the differential diagnosis should be rewritten as to include the differences to Kwanasaurus taxon by taxon.
Line 449: replace “dinosauromorph elements” by “dinosauromorphs”
Line 450: delete “any” before “elements”
Line 457: delete the entire line as it provides no relevant information in the given context.
Line 460: replace “often” by “taxa often fall into”.
Line 472: delete “a” after “least”
Lines 473-474: replace “that is subject to potential falsification with the discovery of associated material, subject to revision if more complete specimens are ever recovered” by “such hypothesis is subject to potential falsification by the discovery of associated material, and subject to revision if more complete specimens are recovered”
Line 479: delete “known from the collection”
Line 480: delete “long”
Line 484: replace “medial flange and their robust form compared to other” by “flange the bone has on its medial surface and their robust form compared to the maxillae of other”
Line 500: include “other” before “Eagle”
Line 522: replace “although” by “and”
Line 528: replace “here” by “in this position”
Lines 529-530: replace “at the base of the ascending process of the maxilla (Bittencourt et al., 2014, p. 191) may be homologous (labeled “pm.ar?” in Fig. 10C)” by “(labeled “pm.ar?” in Fig. 10C) at the base of the ascending process of the maxilla (Bittencourt et al., 2014, p. 191) may be homologous to that concavity”
Line 544: replace “but it is also” by “end”
Line 596: include “main part of the” before “maxilla”
Line 598: delete “on its surface”
Line 604: include “that of” after “to”
Line 666: replace “silesaurid dentary described” by “dentary described for a silesaurid”
Line 667: delete “of the element” and include “silesaurid” after “described”
Line 673: include “in” after “probably” and replace “is” by “of”
Line 727: include “that it” after “indicate”
Line 729: include “that of” after “from”
Line 751: replace “was” by “is”
Line 766 and elsewhere: it is really not clear to me that the “Meckelian groove extends to the anterior tip of the dentary”, it could in fact end rostrally where it “narrows to almost nothing beneath the third tooth position”. I believe that a stronger argument should be put forward here indicating why the more rostral grooves are also part of the Meckelian groove and not only grooves related to the dentary symphysis.
Lines 786-787: last sentence is repeated and can be excluded.
Line 789: include “of the maxilla” after “process”
Line 796: replace “tooth” by “teeth”
Line 814: replace “ridges” by “ridge”
Line 844: replace “maxilla” by “maxillae”
Line 845: include “are” after “they”
Line 858: perhaps discuss here if the “canting” stands as an apomorphy uniting Sacisaurus plus Diodorus, as proposed by Kammerer et al. (2012).
Line 927: replace “dentary” by “dentaries”
Line 932: delete “(“ before “while”
Line 938: delete “teeth” before “tooth”
Line 940: delete “patterns” before “pattern”
Lines 942-945: delete the last sentence of the paragraph, it bears no relevant information on the given context.
Line 969: the abbreviations “ect” and “ent” (epicondyles) cannot be used in reference to the condyles (as seems to be the case here). Please modify.
Line 971: delete “here” after “occurs”
Line 984: replace “is probably” by “probably belongs to” and “from” by “in relation to”
Lines 990-991: a horizontally oriented iliac blade is a rather unusual feature. It seems that this part of the bone is not complete even in the best-preserved ilium of Kwanasaurus. It would be nice to see a more extensive explanation of this condition here, or even some measure of uncertainty (if this is the case).
Line 1000 and 1009: “flattened”/”flattens” in which direction?
Line 1052: include “dorsoventrally” before “deep”
Lines 1059-1060: exclude part that compares with Ornithosuchus, or explain why this animal is used only here for comparison.
Line 1076: delete “this” after “blade”
Line 1085: explain why, or provide reference to such, the third sacral vertebra may be the “primordial second”.
Line 1091: Silesaurus has clearly at least three sacral vertebrae, perhaps four (although probably not), but surely not two.
Lines 1104-1116: this part is confused and needs rewriting. First it is said that “two of the following” (the following supposedly being those numbered 1-3 below) features are seen in all silesaurids, then a “fourth” feature appears in the paragraph (lines 1107-1108). Also, the item number 3 is not merely the description of a feature (as it should be and as is the case of items 1-2), but also includes its distribution in Asilisaurus. Please, modify as to have itemised all and only character descriptions.
Line 1120: include “of the” before “shape”
Line 1139: would it be “proximally” instead of “distally”?
Line 1162: “swelling with the distal end of the lesser trochanter” is unclear.
Line 1168: replace “the trochanteric shelf” by “it”
Line 1171: include “is probably” before “highly”
Line 1182: delete “other” after “in”
Line 1183: replace “and” by “which are”
Line 1203 and 2013: could it be “anterior” instead of “behind”?
Line 1204: replace “17B” by “18B”
Line 1235: delete “so that”
Lines 1266-1271: rewrite the paragraph as follows “Sulcimentisauria is proposed as the name for the clade including all silesaurids closer to Silesaurus opolensis than to Asilisaurus kongwe (see Systematic Paleontology). Within Sulcimentisauria, Eucoelophysis baldwini and Diodorus scytobrachion are consecutive sister taxa to other sulcimentisaurians, which includes a clade comprising Lutungutali sitwensis and Ignotosaurus fragilis in polytomy with Silesaurus opolensis, Sacisaurus agudoensis, and Kwanasaurus williamparkeri (Figure 23). None of the phylogenetic relationships within Sulcimentisauria.”
Line 1305: “so few approach being complete that little” is unclear.
Line 1383: An Anisian taxon cannot be younger that a Ladnian taxon…
Line 1397: include “the groups” after “and”
Line 1403: replace “now” by “also”; perhaps “major” is too a strong works considering the comment that follows in line 1406: “they are generally rare”.
Line 1404: delete “in” after “and”
Line 1413: non-italicize “and”
Line 1424: replace “represents” by “bear”
Line 1425: replace “the” by “their”

---

## Round 0.2 · accepted · Accept

Thank you for your careful revision of the manuscript. I have read the new version and see no need for additional review.